# THINKING WITH TIME SERIES: INTERLEAVED DEEP THINKING FOR ENHANCED TIME SERIES REASONING

## ABSTRACT

Understanding and reasoning with time series is an important yet unsolved challenge for multimodal large language models (MLLMs). Current time series MLLMs (TS-MLLMs) often struggle with complex tasks due to their overly simplified reasoning process. In this work, we argue that deep thinking is essential for comprehensively understanding and effectively reasoning over time series. We present *ThinkTime*, the first TS-MLLM that supports Interleaved Time series Chain-of-Thought (iTCoT) with integrated tool calls. In iTCoT, the reasoning process is interleaved with tool calls, allowing the model to dynamically incorporate information from time series slices into its thought process. To enable comprehensive analysis, the model introduces two fundamental operations, *slice* and *compare*, which are designed to observe detailed and correlation features. To achieve this, we design a two-stage training process and propose a task-specific training data construction method based on synthetic data. In the supervised fine-tuning stage, we use an iTCoT dataset to teach the model how to integrate tool responses with reasoning processes. Then, in the reinforcement learning stage, we implement an RL training framework for TS-MLLMs that supports iTCoT, improving the model's reasoning and tool-use abilities. Experiments conducted on a wide range of real-world time series demonstrate that ThinkTime achieves substantial improvements in reasoning tasks while maintaining high alignment between time series and text descriptions.

## 1 INTRODUCTION

With the demonstrated capabilities of Multimodal LLMs (MLLMs) in the time series domain, an increasing number of studies (Jin et al., 2023; Wang et al., 2025a; Gruver et al., 2024) have adopted these models for time series analysis. Among various tasks, time series understanding and reasoning have received significant attention (Xie et al., 2025; Wang et al., 2025c; Liu et al., 2025a; Kong et al., 2025a). They take text and time series data as input, and answer user questions in text form through multimodal understanding and reasoning. However, Time Series MLLMs (TS-MLLMs) are still in the early stage of exploration, especially those involving complex reasoning tasks (Kong et al., 2025b; Jin et al., 2024). To address this gap, several recent works have introduced *deep thinking* (*i.e.*, long-CoT) into TS-MLLM for other applications such as forecasting (Luo et al., 2025). While these approaches explore the advantages of deep thinking for time series, deep thinking for time series understanding and reasoning has not been studied. Furthermore, existing studies that combine deep thinking with TS-MLLMs are limited to *text-only* chain-of-thought (CoT) approaches, restricting their ability to perform detailed reasoning about time series.

To enable *multimodal* deep thinking, OpenAI introduced "Thinking with Image" in *o3* (OpenAI, c). In deep thinking, the text content is *interleaved* with multiple cropped parts of the image to reason details in different regions of the image through several *tool calls*. This mimics how humans reason over images by focusing on different elements and their relationships, and recent studies have shown its effectiveness in complex reasoning tasks (Zheng et al., 2025; Singh et al., 2025). Similarly, understanding and reasoning over time series data also relies heavily on observing and analyzing different details, indicating that this approach is applicable to time series. However, several challenges must be addressed to realize this. First, unlike images, time series lack standardized reasoning tasks to set model optimization objectives (Chow et al., 2024; Merrill et al., 2024; Sen et al., 2025), which are essential for leveraging reinforcement learning (RL) to realize enhanced

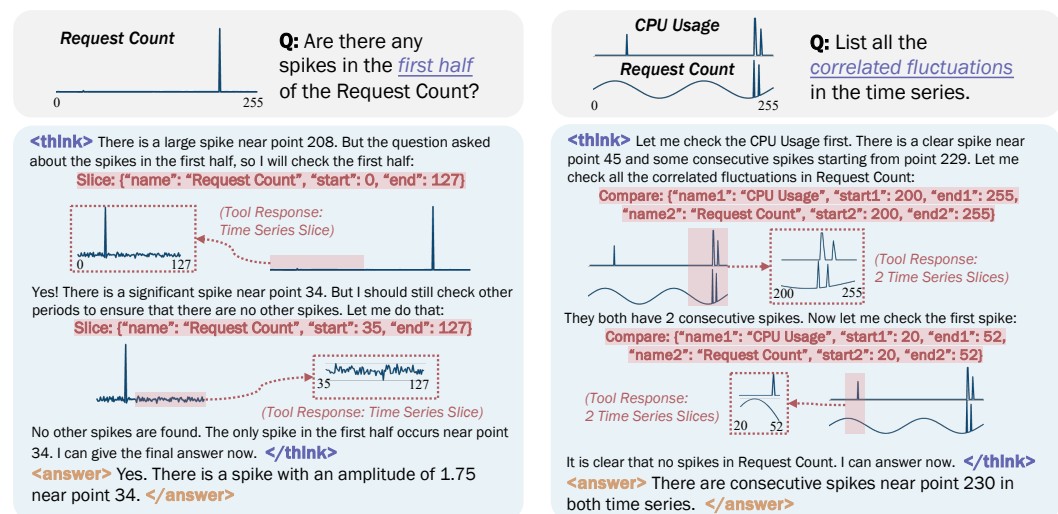

Figure 1: Example of the iTCoT paradigm to realize "Thinking with Time Series". *Slice* and *Compare* are interleaved with text output in the thinking process. LLMs conduct detailed reasoning by analyzing diverse segments from different series within MTS.

deep thinking. Furthermore, there are almost no training data for time series reasoning to support both supervised fine-tuning (SFT) and RL (Luo et al., 2025; Xie et al., 2025). In addition, the absence of open-source RL frameworks for TS-MLLMs makes it highly challenging to implement *interleaved deep thinking* for time series.

In this paper, we propose *ThinkTime*, the first approach that explores **i**nterleaved **T**ime series **C**hain-**o**f-**T**hought (**iTCoT**) to tackle complex understanding and reasoning tasks on time series. Building on existing work and inspired by the way humans analyze time series, we define two fundamental operations essential for iTCoT: *slice* and *compare* (see Figure 1). The *slice* operation extracts a specific segment from one of the time series, enabling detailed analysis of features within a particular region. The *compare* operation extracts two different segments from either the same or different time series, allowing the MLLM to analyze autocorrelation or cross-correlation features. These operations allow the MLLM to adapt to multi-scale time series and focus on critical regions during deep thinking, thereby improving its ability for complex time series understanding and reasoning.

In terms of training, we observe that existing TS-MLLMs (*e.g.*, ChatTS(Xie et al., 2025)) do not natively support deep thinking or tool calling in iTCoT. To address this limitation, we design a comprehensive data pipeline, from a warmup SFT stage to an RL stage, which enables TS-MLLMs to perform multimodal deep thinking. When using existing LLMs to generate iTCoT for warmup SFT, their inability to produce accurate deep thinking with tool calls motivates us to design a task-specific generation method that enables TS-MLLMs to quickly adapt to time series reasoning. For RL, we construct a multi-task RLVR (Guo et al., 2025) dataset based on synthetic data. We further implement the first framework for TS-MLLMs that supports iTCoT for RL training with DAPO (Yu et al., 2025) based on the TRL framework (von Werra et al., 2020). Moreover, to ensure comprehensive evaluation, we curate a collection of datasets comprising various reasoning and alignment tasks, primarily consisting of *real-world time series*. **Our contributions are summarized as follows:**

- We propose the paradigm of "Thinking with Time Series", which introduces interleaved Time Series Chain-of-Thought (iTCoT) for complex time series understanding and reasoning. We define two fundamental operations: *slice* and *compare* that enable models to analyze local features and cross-series relationships more effectively.

- We propose a comprehensive data pipeline to support iTCoT, including task-specific SFT dataset generation and multi-task RLVR dataset construction for RL. We also implement the first training framework for TS-MLLMs with iTCoT to realize DAPO training.

- Based on the proposed data pipeline and training framework, we train ThinkTime, the first TS-MLLM supporting iTCoT for enhanced time series reasoning.

- We curate evaluation datasets covering 11 categories of real-world time series tasks, including both alignment and reasoning. ThinkTime achieves superior performance over existing text/vision/agent/TS methods based on SOTA LLMs.

## 2 RELATED WORK

**Time Series Multimodal LLMs.** The rapid progress of multimodal large language models (MLLMs) has shown the effectiveness of aligning language models with diverse modalities (Bai et al., 2023; Maaz et al., 2023). Motivated by these advances, researchers have begun to explore the integration of LLMs with time series data. Several studies adapt pretrained LLMs for downstream tasks, including forecasting and anomaly detection (Chang et al., 2023; Liu et al., 2024c; Jin et al., 2023), while others leverage VLM architectures to treat time series as images (Chen et al., 2024; Zhuang et al., 2024; Liu et al., 2024b). Although recent attempts have combined LLMs with time series, most methods rely on task-specific adaptations or heuristic pipelines, often restricted to retrieval-augmented generation (RAG) (Lewis et al., 2020) or agent-based reasoning (Yao et al., 2022; Zhou et al., 2023). These approaches lack systematic language–time series alignment and struggle to capture the deeper reasoning capabilities required for general-purpose time series understanding and reasoning (Kong et al., 2025a;b; Jin et al., 2024). To address this gap, recent work based on TS-MLLMs (Xie et al., 2025; Wang et al., 2025a;c; Quinlan et al., 2025) introduces synthetic datasets or LLM-generated datasets, providing a solution for aligning LLMs with time series modalities. Some recent work (Liu et al., 2025b) also employs deep thinking based on a text-only manner for forecasting. However, current models focusing on reasoning tasks of Time Series MLLMs are still in the early exploratory stage (Merrill et al., 2024; Sen et al., 2025), without enough capabilities of multimodal deep thinking.

**Multimodal Deep Thinking.** Since OpenAI's *o1* model introduced deep thinking for text reasoning (OpenAI, b). RL has been widely used to enhance reasoning in LLMs (Gao et al., 2024; Team et al., 2025; OpenAI, b). Methods such as GRPO (Guo et al., 2025) and DAPO (Yu et al., 2025) improve training stability and efficiency by refining their training rewards. *o3* extended this to interleaved multimodal thinking (OpenAI, c). Recent work (Zheng et al., 2025; Chen et al., 2025; Wang et al., 2025b) reproduced this idea and trained VLMs with agentic RL (Singh et al., 2025). However, these advances remain focused on text or vision, and there is still no framework that supports multimodal deep thinking with time series.

## 3 METHOD

In this section, we first introduce the idea of "Thinking with Time Series" in Section 3.1. Then, we introduce the model design and implementation in Section 3.2 and its training process in Section 3.4. Finally, we introduce the data generation process in Section 3.3.

### 3.1 THINKTIME

ThinkTime is a *Time Series MLLM* (TS-MLLM) that takes the native *multivariate time series (MTS)* array with a text question as input and text as output for reasoning and understanding tasks. Therefore, it is structurally different from existing VLMs (Liu et al., 2024a). Similar to multimodal deep thinking models such as *o3* (OpenAI, c), the proposed ThinkTime introduces interleaved multimodal deep thinking into time series MLLMs for the first time. To achieve this, we need to equip TS-MLLM with: (1) *Deep Thinking Capability*: the model can perform deep thinking based on time series and questions before generating answers to solve complex reasoning problems; (2) *Tool Call Capability*: during the deep thinking process, the model actively invokes predefined tools to explore details or verify results, ensuring accuracy of analysis with interleaved time series chain-of-thought (iTCoT). To realize these two capabilities, we applied: (1) Warm-Up SFT, which gives the model an initial ability of "Thinking with Time Series"; (2) RL, which further enables the model to integrate tool calls into deep thinking, significantly improving reasoning ability.

Specifically, during the deep thinking process, we invoke the two types of tools mentioned earlier, *slice* and *compare*, in the form of tool calls. For an input MTS: $\mathbf{X} \in \mathbb{R}^{N \times T}$, slice and compare perform the following operations:

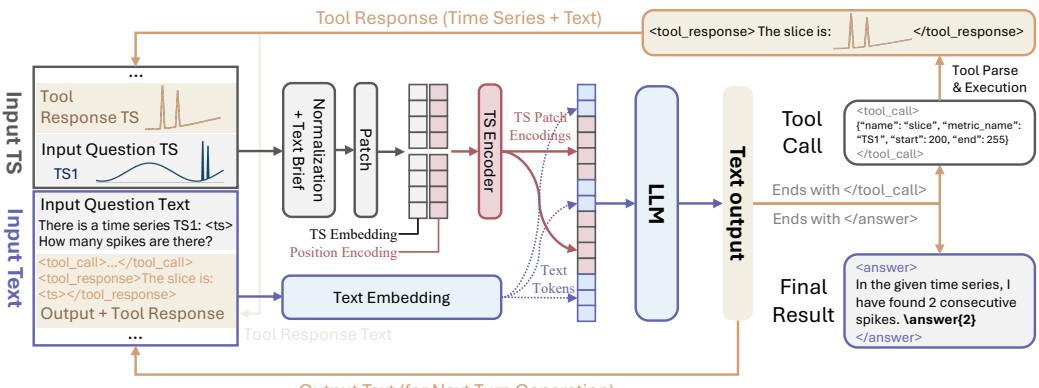

Figure 2: Model structure of ThinkTime. It takes both time series and text as input, with time series patches encoded and concatenated with text embeddings before being fed into the LLM. During iTCoT, the model generates tool calls, which are parsed and executed in a multi-turn process. The resulting tool responses are integrated with the text output until the final answer is produced.

- **Slice.** Slice selects a time series segment $\mathbf{X}^i_{[t1:t2]}$, where $i \in \{1, \cdots, N\}$. Similar to the "image cropping" operation, the LLM invokes this operation when it needs to conduct a detailed analysis on a specific segment of a designated time series, or when it needs to verify a certain part of the segment.

- **Compare.** Compare is essentially similar to performing two slice operations. Similar to deep thinking with images, time series analysis relies heavily on comparative analysis to extract autocorrelation and cross-correlation features. It can be achieved by comparing two segments by invoking the slice tool twice: $\mathbf{X}^i_{[t1:t2]}$ and $\mathbf{X}^j_{[t3:t4]}$. For convenience, we merge them into a single *compare* operation. For autocorrelation analysis (such as periodicity analysis and local fluctuation analysis), the LLM compares different time slices within the same time series (*i.e.*, $i = j$). For cross-correlation analysis, the LLM compares the same slices across different time series (*i.e.*, $t1 = t3, t2 = t4$).

## 3.2 MODEL DESIGN

The overall structure of the proposed ThinkTime is shown in Figure 2. To achieve iTCoT, following common practice in VLMs (Zheng et al., 2025), the deep thinking–tool calling process adopts a multi-turn dialogue style. After the model generates a piece of content, its output is examined to decide whether to call the tool (ending with `</tool_call>`) or whether it has already completed the full response (ending with `</answer>`). The specific format of tool calls is JSON, which includes the tool name, the target time series, and the start and end time points. The returned result of the tool call is appended to the previous round of input for the next round of generation, and this continues until no further tool calls are made. More examples of iTCoT are demonstrated in Figure 1 and Appendix A.

To enable multimodal input of MTS and text, ThinkTime adopts a native multimodal input method. First, to preserve the original information as much as possible, following ChatTS (Xie et al., 2025), we apply 0–1 normalization to the time series while also generating a *text brief* that retains its original values and statistical information (*e.g.*, `[offset=-12.92|scaling=19.11|length=50|max=12.23|min=12.33]`). Each time series is divided into fixed-size patches, which are combined with learnable position embeddings and encoded by MLP layers to obtain *TS Patch Encodings*. These encodings are concatenated with text embeddings according to their original positions and fed into the LLM backbone. To achieve iTCoT with detailed slice features, for each time series slice from a tool response, we perform *per-slice* normalization to fully preserve the details of each slice.

## 3.3 TRAINING DATA

Existing TS-MLLMs cannot perform deep thinking or tool calling. We therefore use WarmUp-SFT data to provide initial iTCoT capabilities and further strengthen them through RLVR (Reinforcement Learning with Verifiable Rewards) (Guo et al., 2025). However, existing time series + text data are *scarce*, and open-source data are *insufficient* to support the construction of datasets required for these two stages. Inspired by ChatTS, we propose to construct Warm-Up SFT and RLVR datasets based on **synthetic time series generator** to get diverse and accurate time series + text pairs for training.

**Warm-Up SFT Data.** The WarmUp-SFT data is used to guide LLMs to acquire initial deep thinking and tool call capabilities. Therefore, we need to construct iTCoT samples within Deep Thinking for model training. We use the synthetic time series generator proposed in ChatTS to generate time series and the corresponding attributes. Based on this, we constructed a dataset with various alignment tasks using templates. Specifically, given a time series, LLMs are required to output basic information such as trend, seasonality, and correlation, without the need to construct complex reasoning tasks. The detailed data construction process is provided in Appendix D.2.

To obtain CoT with tool calls, directly prompting an LLM to generate iTCoT often leads to poor dataset quality, with duplicate or meaningless tool calls. A key reason is that existing LLMs lack the ability to select appropriate time series tools for different tasks or to coordinate them with deep reasoning. This limitation causes overthinking and excessive tool calls, rendering the generated datasets unusable. To address this, we design a *task-specific* process for iTCoT generation. During generation, we provide the LLM with different CoT process prompts according to the type of the pre-defined task. In this way, we concatenate the interleaved CoT by LLMs with the tool responses and format them into a multi-turn dialogue dataset. Detailed prompts are given in the Appendix D.2.

**RLVR Data.** The RLVR data consists of two categories: alignment tasks and reasoning tasks. The data format for alignment tasks is the same as in Warm-Up SFT, except that iTCoT is no longer required. For reasoning tasks, task-specific generation processes are not required, which allows us to use LLMs to generate a wide variety of QA pairs. During generation, we apply prompts to guide the LLM in producing different types of reasoning questions based on synthetic time series and their text descriptions. To make the rewards verifiable, the answers were restricted to multiple-choice, true/false, and numerical questions. Finally, another LLM is used to check whether each QA pair was accurate and whether the answer was non-trivial with respect to the time series description. More details of data generation are provided in Appendix D.3.

## 3.4 MODEL TRAINING

To achieve iTCoT, we perform WarmUp SFT and RL based on ChatTS-14B (Xie et al., 2025). We first apply WarmUp SFT to provide the model with initial iTCoT capability based on the alignment dataset with iTCoT. We further applied RLVR to strengthen the model in complex reasoning.

**Warm-Up SFT.** In the warm-up stage, we train the model with SFT using a multi-turn dialogue format to achieve the interleaved CoT. Each training sample is organized as a dialogue between the model and the tool's responses. The dialogue alternates between model outputs and tool responses until the final answer is generated. This design helps the model to gradually learn how to interleave reasoning with slice and compare operations. The use of synthetic time series and alignment tasks gives the model basic capability of iTCoT. The warm-up SFT stage provides the model with initial skills in deep thinking with time series, but without reinforcement on reasoning quality.

**Reinforcement Learning.** Following warm-up SFT, we employ reinforcement learning to further improve reasoning ability, using DAPO (Yu et al., 2025) as the optimization algorithm. To support this setting, the TRL framework (von Werra et al., 2020) is modified and adapted to enable iTCoT under DAPO. Following agentic RL (Singh et al., 2025; Zheng et al., 2025), we extend the rollout formulation to time series by treating tool responses as external observation tokens that are interleaved with text tokens and fed back into the model during iTCoT. Each rollout involves multi-turn generation, where tool calls and their responses are repeatedly incorporated until the final answer is produced. The reward function is carefully designed to balance correctness, formatting, and reasoning length. Specifically, the total reward is defined as:

$$R(\tau) = w_{acc} \cdot R_{acc}(\tau) + w_{format} \cdot R_{format}(\tau) + w_{len} \cdot R_{len}(\tau), \tag{1}$$

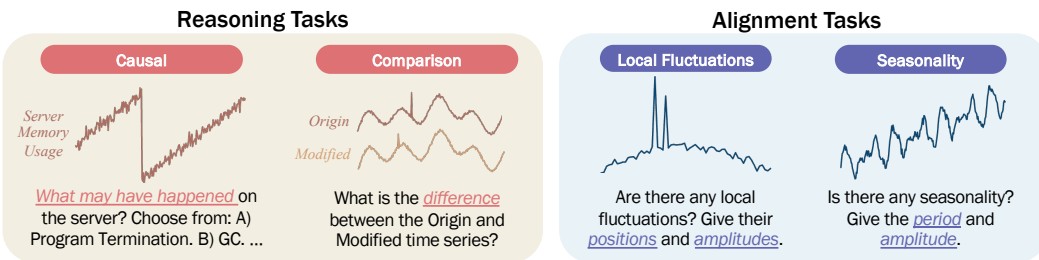

Figure 3: Examples of **reasoning** and **alignment** tasks. Reasoning tasks require combining time series features with the question text for reasoning. It requires both multimodal recognition and reasoning capabilities. Alignment tasks focus more on the accuracy of multimodal recognition.

where $R_{acc}(\tau)$ is the accuracy reward that measures whether the final answer is correct, $R_{format}(\tau)$ is the formatting reward that penalizes unstructured outputs, $R_{len}(\tau)$ is the thinking length reward, and $w_{...}$ are the weights of each reward. More details can be found in Appendix C.1. By combining these three components, the total reward encourages the model to generate correct, well-structured, and sufficiently detailed reasoning trajectories. With this reward design, reinforcement learning improves the model's capability of deep reasoning and tool using with time series.

## 4 EVALUATION

### 4.1 TASKS AND DATASETS

To comprehensively evaluate the performance of ThinkTime on complex time series problems, we collected and annotated a large number of datasets from different domains (see Figure 3). Most of the evaluation questions use **real-world time series**.

**Reasoning Tasks.** Following existing research (Xie et al., 2025; Wang et al., 2025c; Sen et al., 2025), we use five reasoning tasks for comprehensive evaluation: pattern recognition, numerical judgment, calculation, causal, and comparison. These tasks require the model to accurately identify both categorical and numerical features of time series (e.g., trend, seasonality, fluctuation) and perform reasoning based on the given conditions in the questions to produce answers. The detailed task definitions and data collection methods for each category are provided in Appendix B.1. For the first four categories, we manually collected and constructed a set of questions from real-world time series. For the comparison task, following Xie et al. (2025), we used MCQ2 (Merrill et al., 2024), a third-party dataset that contains synthetic time series and LLM-generated questions. Refer to Appendix B.3 for details of dataset construction.

**Alignment Tasks.** To more comprehensively evaluate the capabilities of ThinkTime, we further used alignment tasks to ensure that the model maintains good alignment between time series and text while gaining stronger reasoning ability. For fairness, we used a dataset from Xie et al. (2025) that contains only real-world time series with human annotations for evaluation. This dataset includes six subtasks, including trend, season, noise, local, correlation, and cluster, covering the evaluation of both UTS and MTS. Details are provided in Appendix B.1.

### 4.2 SETUPS

**Baselines.** Following existing related studies (Xie et al., 2025; Sen et al., 2025), our baselines cover four categories of models: Text, Vision, Agent-Based, and TS-MLLM-Based. For the first three categories, we selected GPT-5 series (OpenAI, a), Claude-Sonnet-4 (Anthropic), and Doubao-1.6-Thinking (Seed) with text-only deep thinking capabilities. We also evaluate o3 (OpenAI, c) for the vision-based method to compare its multimodal deep thinking with the proposed iTCoT with native time series input. Refer to Appendix B.4 for details of baselines.

**Implementation Details.** ThinkTime is trained based on ChatTS-14B (Xie et al., 2025) model (which is fine-tuned based on Qwen2.5-14B-Instruct (qwe, 2024)). For the evaluation of text-,

Table 1: Comparison of different models in **reasoning tasks**. All metrics are the higher, the better. Best results are represented in **red** and second-best results are represented in blue.

| Type | Model | Pattern | | Numerical | | Calculation | | Causal | | Comparison | Average | |
|---|---|---|---|---|---|---|---|---|---|---|---|---|
| | Task | UTS | MTS | UTS | MTS | UTS | MTS | UTS | MTS | MTS | UTS | MTS |
| Text | GPT-4o | 0.369 | 0.556 | 0.660 | 0.595 | 0.480 | 0.434 | 0.510 | 0.536 | 0.470 | 0.505 | 0.518 |
| | GPT-5-mini | 0.476 | 0.667 | 0.755 | 0.676 | 0.532 | 0.581 | 0.612 | 0.821 | 0.580 | 0.594 | 0.665 |
| | GPT-5 | 0.631 | 0.630 | 0.868 | 0.595 | 0.582 | 0.623 | 0.633 | 0.786 | 0.590 | 0.679 | 0.645 |
| | Claude-Sonnet-4 | 0.553 | 0.556 | 0.849 | 0.676 | 0.579 | 0.660 | 0.796 | 0.821 | 0.630 | 0.694 | 0.669 |
| | Doubao-1.6 | 0.369 | 0.185 | 0.472 | 0.432 | 0.417 | 0.351 | 0.429 | 0.429 | 0.470 | 0.422 | 0.373 |
| | QWen2.5-14B | 0.340 | 0.259 | 0.396 | 0.568 | 0.326 | 0.477 | 0.408 | 0.536 | 0.320 | 0.368 | 0.432 |
| Vision | GPT-4o | 0.670 | 0.482 | 0.642 | 0.622 | 0.515 | 0.566 | 0.837 | 0.857 | 0.490 | 0.666 | 0.603 |
| | GPT-5-mini | 0.670 | 0.556 | 0.736 | 0.649 | 0.580 | 0.779 | 0.796 | 0.929 | 0.490 | 0.696 | 0.681 |
| | GPT-5 | 0.767 | 0.630 | 0.755 | 0.811 | 0.661 | 0.760 | 0.776 | 0.963 | 0.610 | 0.740 | 0.755 |
| | Claude-Sonnet-4 | 0.495 | 0.556 | 0.736 | 0.676 | 0.532 | 0.750 | 0.796 | 0.750 | 0.590 | 0.640 | 0.664 |
| | Doubao-1.6 | 0.495 | 0.482 | 0.679 | 0.568 | 0.606 | 0.826 | 0.633 | 0.714 | 0.610 | 0.603 | 0.640 |
| | o3 | 0.757 | 0.630 | 0.830 | 0.595 | 0.566 | 0.756 | 0.796 | 0.821 | 0.450 | 0.737 | 0.650 |
| Agent | GPT-4o | 0.495 | 0.519 | 0.623 | 0.487 | 0.181 | 0.188 | 0.245 | 0.571 | 0.470 | 0.386 | 0.447 |
| | GPT-5-mini | 0.379 | 0.482 | 0.698 | 0.649 | 0.169 | 0.275 | 0.408 | 0.714 | 0.420 | 0.414 | 0.508 |
| | GPT-5 | 0.485 | 0.667 | 0.566 | 0.595 | 0.227 | 0.315 | 0.347 | 0.607 | 0.540 | 0.406 | 0.545 |
| | Claude-Sonnet-4 | 0.320 | 0.482 | 0.547 | 0.595 | 0.293 | 0.213 | 0.429 | 0.464 | 0.570 | 0.397 | 0.465 |
| | Doubao-1.6 | 0.359 | 0.593 | 0.604 | 0.405 | 0.170 | 0.184 | 0.286 | 0.464 | 0.540 | 0.355 | 0.437 |
| TS | ChatTS-14B | 0.369 | 0.519 | 0.604 | 0.405 | 0.507 | 0.397 | 0.755 | 0.857 | 0.600 | 0.559 | 0.556 |
| | **ThinkTime-14B** | **0.903** | **0.815** | **0.887** | **0.875** | **0.794** | **0.850** | **0.918** | 0.892 | **0.680** | **0.876** | **0.822** |

Table 2: Comparison of different models in **alignment tasks**. "Cate." and "Num." denotes categorical and numerical tasks respectively.

| Type | Model | Trend | | Season | | Noise | | Local | | Corr. | Clus. | Overall | |
|---|---|---|---|---|---|---|---|---|---|---|---|---|---|
| | Task | Cate. | Num. | Cate. | Num. | Cate. | Num. | Cate. | Num. | Cate. | Cate. | Cate. | Num. |
| Text | GPT-4o | 0.585 | 0.882 | 0.811 | 0.768 | 0.905 | 0.153 | 0.379 | 0.256 | 0.476 | 0.333 | 0.542 | 0.371 |
| | GPT-5-mini | 0.585 | 0.739 | 0.378 | 0.205 | 0.929 | 0.096 | 0.513 | 0.427 | 0.643 | 0.262 | 0.579 | 0.452 |
| | GPT-5 | 0.561 | 0.773 | 0.703 | 0.828 | 0.952 | 0.252 | 0.514 | 0.454 | 0.690 | 0.337 | 0.636 | 0.498 |
| | Claude-Sonnet-4 | 0.610 | 0.790 | 0.865 | 0.791 | 0.881 | 0.283 | 0.655 | 0.622 | 0.762 | 0.533 | 0.737 | 0.621 |
| | Doubao-1.6 | 0.659 | 0.786 | 0.676 | 0.000 | 0.929 | 0.208 | 0.434 | 0.420 | 0.548 | 0.412 | 0.606 | 0.467 |
| | QWen2.5-14B | 0.707 | 0.709 | 0.622 | 0.205 | 0.833 | 0.231 | 0.137 | 0.099 | 0.571 | 0.349 | 0.464 | 0.241 |
| Vision | GPT-4o | 0.659 | 0.613 | 0.811 | 0.559 | 0.810 | 0.248 | 0.537 | 0.414 | 0.476 | 0.480 | 0.609 | 0.436 |
| | GPT-5-mini | 0.732 | 0.678 | 0.784 | 0.000 | 0.952 | 0.188 | 0.853 | 0.706 | 0.893 | 0.706 | 0.820 | 0.642 |
| | GPT-5 | 0.732 | 0.669 | 0.973 | 0.000 | 0.643 | 0.496 | 0.916 | 0.754 | 0.825 | 0.702 | 0.809 | 0.706 |
| | Claude-Sonnet-4 | 0.683 | 0.615 | 0.946 | 0.000 | 0.714 | 0.189 | 0.688 | 0.638 | 0.786 | 0.571 | 0.739 | 0.583 |
| | Doubao-1.6 | 0.854 | 0.616 | 0.784 | 0.000 | 0.833 | 0.360 | 0.702 | 0.572 | 0.857 | 0.648 | 0.773 | 0.555 |
| | o3 | 0.732 | 0.613 | 0.946 | 0.000 | 0.833 | 0.350 | 0.832 | 0.744 | 0.857 | 0.654 | 0.834 | 0.673 |
| Agent | GPT-4o | 0.610 | 0.501 | 0.432 | 0.205 | 0.667 | 0.201 | 0.242 | 0.184 | 0.357 | 0.330 | 0.404 | 0.248 |
| | GPT-5-mini | 0.195 | 0.185 | 0.081 | 0.205 | 0.738 | 0.397 | 0.299 | 0.213 | 0.095 | 0.274 | 0.336 | 0.226 |
| | GPT-5 | 0.195 | 0.152 | 0.054 | 0.193 | 0.786 | 0.346 | 0.398 | 0.238 | 0.476 | 0.283 | 0.411 | 0.232 |
| | Claude-Sonnet-4 | 0.390 | 0.155 | 0.162 | 0.220 | 0.762 | 0.160 | 0.397 | 0.205 | 0.357 | 0.421 | 0.454 | 0.191 |
| | Doubao-1.6 | 0.561 | 0.196 | 0.027 | 0.205 | 0.667 | 0.374 | 0.108 | 0.057 | 0.262 | 0.447 | 0.307 | 0.117 |
| TS | ChatTS-14B | **0.927** | 0.874 | 0.973 | **0.849** | 0.857 | 0.511 | 0.895 | 0.805 | 0.905 | 0.782 | 0.889 | 0.788 |
| | **ThinkTime-14B** | 0.854 | **0.926** | **1.000** | 0.700 | **0.952** | **0.636** | **0.937** | **0.846** | **0.912** | **0.790** | **0.901** | **0.838** |

vision-, and agent-based models, we referred to the code implementation in ChatTS and further adapted it to the reasoning benchmark. Refer to Appendix C for more training details.

### 4.3 MAIN RESULTS

Table 1 shows that ThinkTime achieves clear improvements in reasoning tasks across both UTS and MTS problems. Compared with strong text baselines such as GPT-5, and the TS baseline ChatTS-14B, our model obtains much higher overall accuracy. The improvements are consistent across almost all reasoning categories, with 13.6% overall improvements compared with all the SOTA models on reasoning tasks. This demonstrates that multimodal deep thinking with slice and compare operations allows the model to capture detailed time series dependencies and conduct more accurate logical reasoning for details. ThinkTime shows significant improvement over its base model (ChatTS-14B), which confirms that reasoning is a key strength of it.

Table 2 further evaluates the performance in alignment tasks. It can be found that the proposed ThinkTime maintains strong performance on almost all the alignment tasks, outperforming strong baselines including GPT-5 and ChatTS. This improvement is exciting since iTCoT is mainly de-

Table 3: Ablation studies on **reasoning tasks**.

| Model | Pattern | | Numerical | | Calculation | | Causal | | Comparison | Average | |
|---|---|---|---|---|---|---|---|---|---|---|---|
| Task | UTS | MTS | UTS | MTS | UTS | MTS | UTS | MTS | MTS | UTS | MTS |
| ChatTS-14B | 0.369 | 0.519 | 0.604 | 0.405 | 0.507 | 0.397 | 0.755 | 0.857 | 0.600 | 0.559 | 0.556 |
| w/o Tool Use | 0.825 | 0.704 | 0.830 | 0.784 | 0.731 | 0.850 | 0.878 | 0.892 | 0.620 | 0.816 | 0.769 |
| w/o RL | 0.845 | 0.815 | 0.774 | 0.622 | 0.777 | 0.842 | 0.898 | 0.892 | 0.630 | 0.824 | 0.760 |
| w/ Workflow (GPT-5) | 0.333 | 0.296 | 0.514 | 0.514 | 0.123 | 0.166 | 0.250 | 0.679 | 0.590 | 0.305 | 0.449 |
| **ThinkTime-14B** | **0.903** | **0.815** | **0.887** | **0.875** | **0.794** | **0.850** | **0.918** | **0.892** | **0.680** | **0.876** | **0.822** |

Table 4: Ablation studies on **alignment tasks**.

| Model | Trend | | Season | | Noise | | Local | | Corr. | Clus. | Overall | |
|---|---|---|---|---|---|---|---|---|---|---|---|---|
| Task | Cate. | Num. | Cate. | Num. | Cate. | Num. | Cate. | Num. | Cate. | Cate. | Cate. | Num. |
| ChatTS-14B | **0.927** | 0.874 | 0.973 | **0.849** | 0.857 | 0.511 | 0.895 | 0.805 | 0.905 | 0.782 | 0.889 | 0.788 |
| w/o Tool Use | 0.756 | **0.968** | 0.973 | 0.635 | 0.929 | 0.461 | 0.821 | 0.740 | 0.860 | 0.704 | 0.826 | 0.755 |
| w/o RL | 0.902 | 0.882 | 0.973 | 0.512 | 0.929 | 0.496 | 0.863 | 0.787 | 0.863 | 0.730 | 0.872 | 0.777 |
| w/ Workflow (GPT-5) | 0.683 | 0.960 | 0.703 | 0.000 | 0.833 | 0.431 | 0.458 | 0.408 | 0.619 | 0.435 | 0.610 | 0.515 |
| **ThinkTime-14B** | 0.854 | 0.926 | **1.000** | 0.700 | **0.952** | **0.636** | **0.937** | **0.846** | **0.912** | **0.790** | **0.901** | **0.838** |

signed for reasoning. This suggests that the introduction of iTCoT can bring better alignment performance for TS-MLLMs because of the capabilities of detailed feature exploration with the tools.

Another interesting finding is that although *o3* achieved good results in some of the evaluation tasks (due to its interleaved CoT with images), its performance is still worse than ThinkTime. The evaluation on alignment tasks shows that *o3* performs well in categorical tasks, but lags behind in numerical tasks (for example, identifying the length of a period, the position of spikes, and their amplitudes). This is because converting time series into images is a feasible approach for time series reasoning tasks, but it causes a significant *loss of details*. Furthermore, *o3*'s interleaved reasoning ability can only process images, and such simple image cropping operations are insufficient for analyzing numerical features that rely on relative positions in the coordinates. In contrast, the proposed ThinkTime adopts native time series input and per-slice normalization, which enables effective analysis of detailed information at different scales of the time series, leading to substantial improvements in both reasoning and alignment tasks.

### 4.4 STUDY OF "THINKING WITH TIME SERIES"

**SFT on only alignment tasks can also improve reasoning through iTCoT.** Although the warm-up stage contains no reasoning datasets, it equips the model with iTCoT (via *slice* and *compare*), which naturally improves reasoning and leads to substantial gains over ChatTS-14B, which is not capable of deep thinking (see the "w/o RL" model in Table 3). This confirms that *thinking with time series* is an effective and transferable capability rather than a task-specific trick.

**iTCoT improves both the reasoning and alignment capabilities.** To show the capability of iTCoT, we removed the tool calls from the Warm-Up SFT dataset and trained a text-only CoT version of ThinkTime (w/o Tool Use) using the same questions and answers. We found that its evaluation results on reasoning and alignment tasks dropped significantly compared to the model equipped with iTCoT. This further demonstrates the important role of iTCoT in different tasks. To further illustrate the performance of iTCoT, we show a case study in Appendix A.

**iTCoT outperforms workflow-style tool calls.** We built a workflow based on GPT-5 that guides the LLM to call the slice and compare tools through prompts. The workflow baseline significantly underperforms both SFT-only and the full model on the reasoning benchmark (Table 3 and Table 4), especially when tasks require multi-step reasoning. Therefore, we believe that embedding iTCoT into the model provides a fundamental improvement compared to the external workflow approach.

**Model with iTCoT is more robust to longer and numerous time series input.** In Figure 4a, we observe clear differences in the number of tool calls across task types and varying numbers of time series. The number of calls also changes during training, which suggests that ThinkTime is

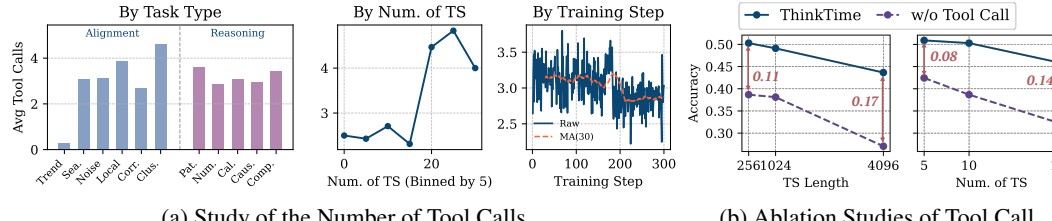

(a) Study of the Number of Tool Calls          (b) Ablation Studies of Tool Call

Figure 4: Study of tool call. ThinkTime calls different numbers of tools depending on the task type, and the number of calls increases with the number of time series. This makes the performance more stable when dealing with a large number of time series and long time series.

able to adaptively adjust its tool use to meet the requirements of different tasks. To further examine this behavior, we compare ThinkTime with its variant without tool calls on MTS alignment tasks in Figure 4b. When either the length or the number of time series increases, ThinkTime maintains significantly stronger performance. These results provide strong evidence that iTCoT plays a key role in improving the robustness of the model.

To conclude, iTCoT is the key to enhancing reasoning in time series. Removing the tool calls results in clear accuracy drops across reasoning categories. These tools provide precise inspection of details and correlations, which bring significantly better reasoning and alignment capabilities to ThinkTime.

### 4.5 STUDY OF REINFORCEMENT LEARNING

To further explore the importance of RL for ThinkTime, we also compare ThinkTime with a variant trained only through SFT (w/o RL). As shown in Table 3, accuracy increases across all categories when RL is applied after WarmUp SFT. This demonstrates that the proposed rewards and DAPO process are effective for correct tool calls and deep thinking. Compared with the SFT-only variant, RLVR reduces errors in complex tasks such as calculation and causal inference, and brings overall performance close to that of the full model. These findings indicate that reinforcement learning plays an essential role in consolidating deep thinking for reasoning tasks.

Reinforcement learning also enhances alignment between time series and text. Table 4 shows small but consistent gains across trend, seasonality, noise, and local subtasks, with particularly strong improvements on numerical tasks where RL helps the model preserve precise values in the output. Although the improvements are less than those in reasoning, the results confirm that RL not only avoids harming alignment but also contributes to stable and balanced performance in analyzing both overall and detailed features of time series. Together, these results demonstrate that the RL training also brings better performance for ThinkTime in both reasoning and alignment tasks.

## 5 CONCLUSION

Understanding and reasoning with time series is challenging for multimodal LLMs due to the scarcity of open-sourced reasoning datasets and training frameworks. In this work, we introduce ThinkTime, the first TS-MLLM that supports iTCoT, which features interleaved time series deep thinking with tool use. Inspired by human strategies for time series analysis, we introduce the *slice* and *compare* operations to support detailed examination and correlation analysis in time series reasoning. We introduce a two-stage pipeline with Warm-Up SFT and RL to train the ThinkTime model through synthetic training datasets. In the Warm-Up SFT stage, we employ a task-specific paradigm for generating multimodal datasets containing iTCoT. In the RL stage, we propose a set of verifiable tasks to construct both reasoning and alignment RLVR datasets. Based on TRL, we built the first DAPO training framework that supports TS-MLLM and added iTCoT support to it. Extensive experiments on real-world time series demonstrate that ThinkTime achieves substantial improvements in reasoning and alignment tasks. Further experiments show that while iTCoT greatly enhances the reasoning and alignment capabilities, it also provides stronger robustness. We believe that the proposed ThinkTime provides a new perspective for improving TS-MLLMs for reasoning.

## 6 ETHICS STATEMENT

This work does not involve human subjects, sensitive personal data, or practices that would raise ethical concerns. We confirm compliance with the ICLR Code of Ethics.

## 7 REPRODUCIBILITY STATEMENT

All the source code (including data generation, warm-up SFT, RL, and evaluation) and datasets are provided in the supplementary material for reproduction. They will also be open-sourced after acceptance of this paper. Please refer to `README.md` in the supplementary material for detailed reproduction steps.

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

# A CASE STUDIES

In this section, we use a series of case studies and comparisons to illustrate how iTCoT works in ThinkTime and its advantages across multiple tasks.

## A.1 COMPARISON BETWEEN THINKTIME AND MODEL WITH TEXT-ONLY CoT

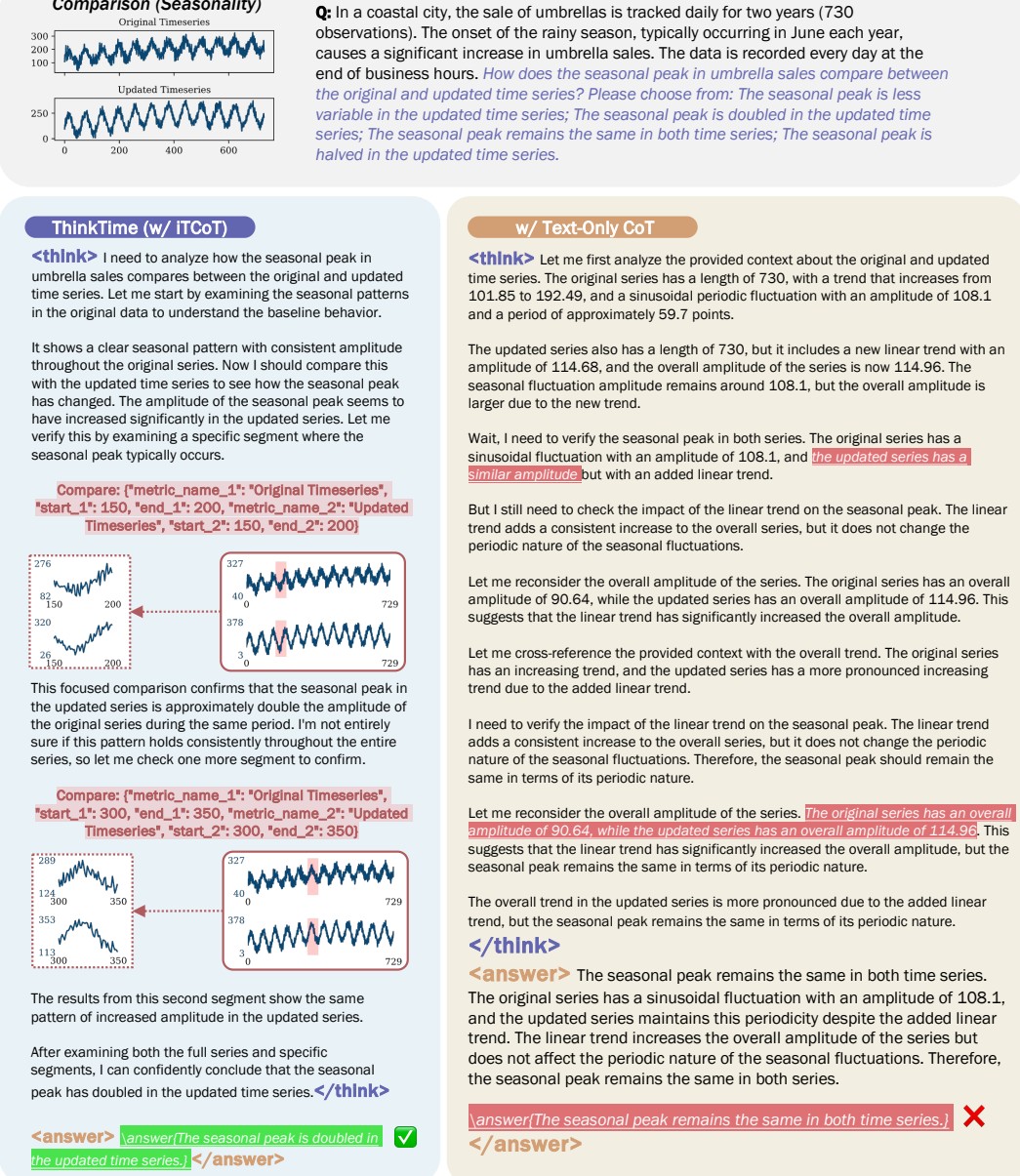

Figure 5: Comparison between ThinkTime and model with text-only CoT in a seasonality case of comparative reasoning.

In Figure 5, ThinkTime accurately analyzed the amplitude changes of each cycle by comparing slices at different positions, and thus obtained the correct result. In contrast, the model with/ text-only CoT incorrectly compared the overall amplitudes of two time series and failed to correct its

initial misanalysis through detailed reasoning in the subsequent steps, which led to an incorrect result.

## A.2 REASONING TASK: NUMERICAL JUDGMENT

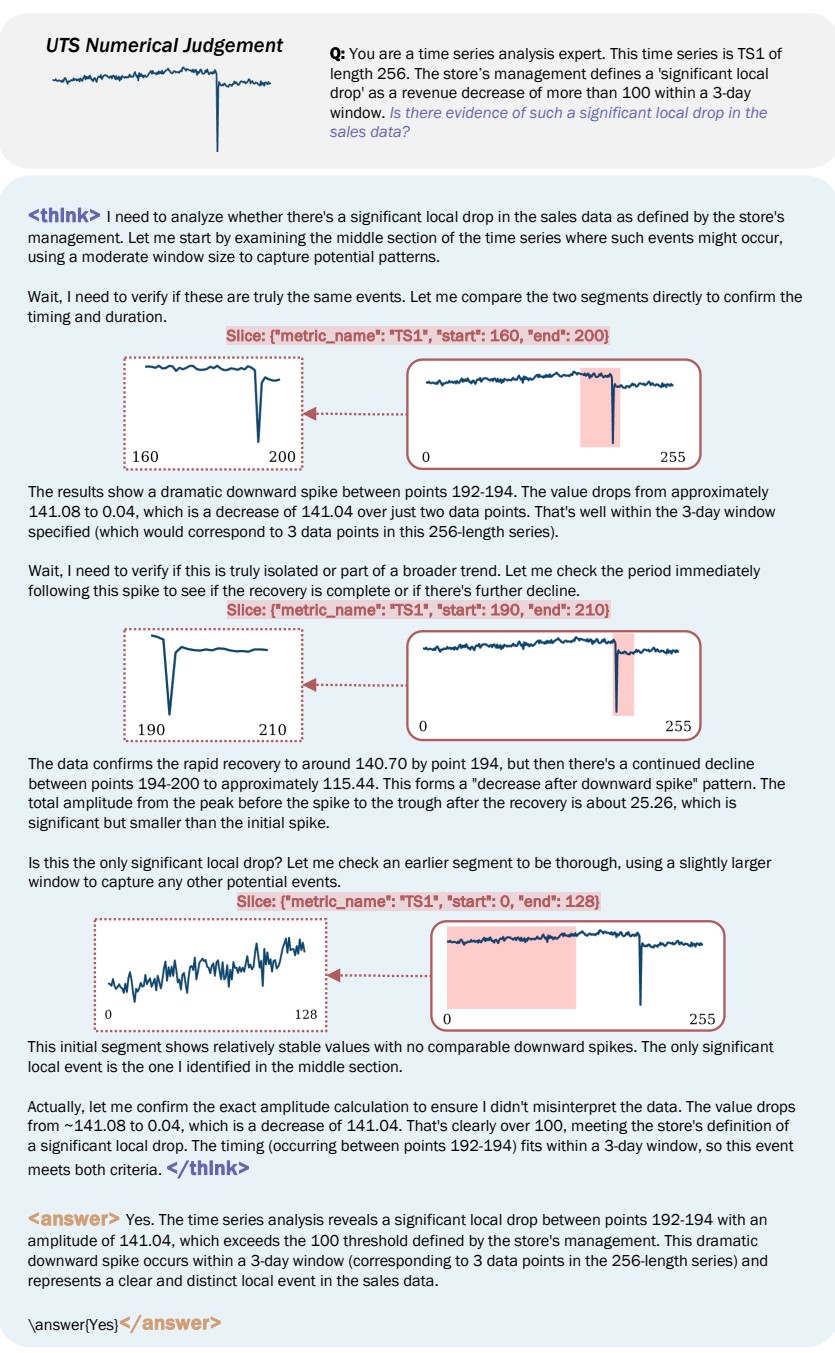

Figure 6: Case study of a numerical judgment reasoning task.

In Figure 6, we present an example of solving a numerical judgement task with iTCoT. In the question, the requirements for numerical judgement are highly detailed, as it requires confirming both the decreased amplitude and the interval. For long sequence inputs, it is very difficult to solve this precisely in one step. ThinkTime used the slicing tool to repeatedly verify the amplitude and position of

the decreasing interval, and ultimately answered the question accurately. This clearly demonstrates the important role of iTCoT in detailed reasoning tasks.

## A.3 REASONING TASK: CALCULATION

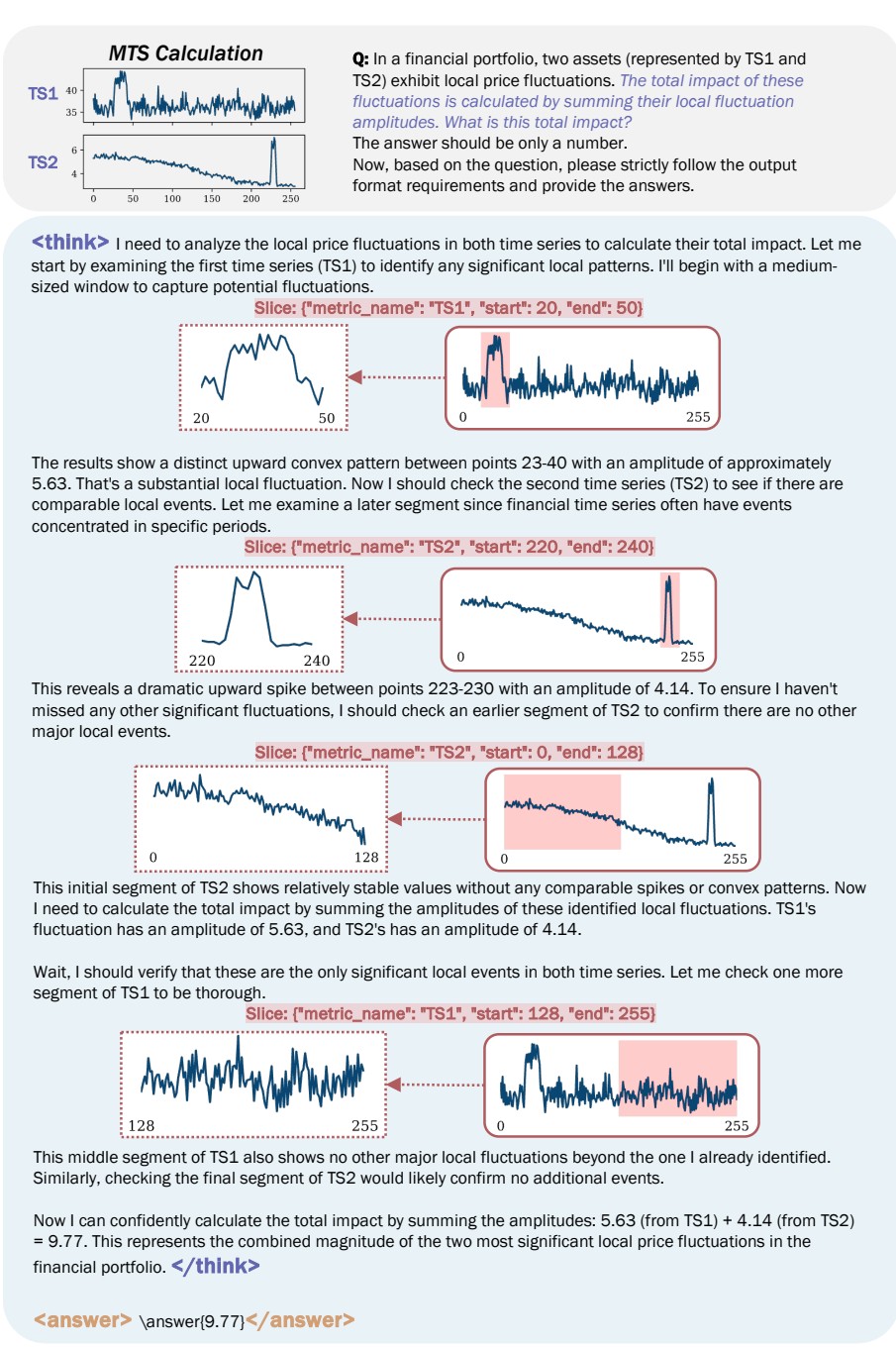

Figure 7: Case study of a calculation reasoning task.

In Figure 7, we present an example of calculation reasoning. The task requires the model to identify all local fluctuations (*e.g.*, spike, convex) in the given MTS and carefully verify their amplitudes. This also requires the model to examine all details to ensure that no possible local fluctuation is missed. As shown, ThinkTime carefully inspected all local fluctuations in the two time series. In

addition, ThinkTime repeatedly invoked extra tools to check other intervals to ensure that nothing was overlooked.

## A.4 REASONING TASK: COMPARISON

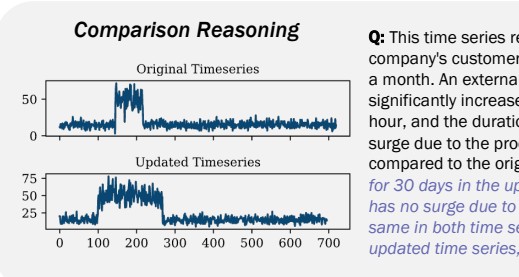

**Comparison Reasoning**

**Q:** This time series represents the number of emails received by a company's customer service department on an hourly basis over a month. An external event, such as a product launch, could significantly increase the volume of emails. The sample rate is 1 hour, and the duration is 30 days. 1. What is the duration of the surge due to the product launch in the updated time series compared to the original? *Please choose from: The surge lasts for 30 days in the updated time series; The updated time series has no surge due to the product launch; The surge duration is the same in both time series; The surge lasts for 7 days in the updated time series, while it lasts 3 days in the original.*

**<think>** I need to analyze the duration of the surge due to the product launch in both the original and updated time series. First, I should examine the original time series to identify when the surge occurs.

There is a significant upward convex pattern starting around point 144, reaching approximately 60.55, then falling back to around 14.00. This forms a distinct surge. To confirm the duration, I'll check the exact start and end points. The surge begins at point 144 and ends around point 175, which is a total of 31 points (approximately 31 hours). In the updated series, I observe a similar upward convex pattern, but with a different amplitude. The surge starts at point 101 and ends around point 105, which is only 4 points (approximately 4 hours).

Wait, I need to verify if these are truly the same events. Let me compare the two segments directly to confirm the timing and duration.

Compare: {"metric_name_1": "Original Timeseries", "start_1": 140, "end_1": 180, "metric_name_2": "Updated Timeseries", "start_2": 95, "end_2": 115}

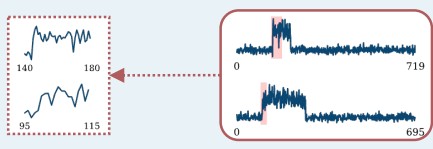

The comparison confirms that while both series exhibit a surge pattern, the duration is significantly shorter in the updated series. The original surge lasts approximately 31 hours, while the updated surge lasts only 4 hours. Is there any possibility I misidentified the start and end points? Let me check a slightly wider window to ensure I didn't miss any part of the surge.

Compare: {"metric_name_1": "Original Timeseries", "start_1": 130, "end_1": 190, "metric_name_2": "Updated Timeseries", "start_2": 90, "end_2": 120}

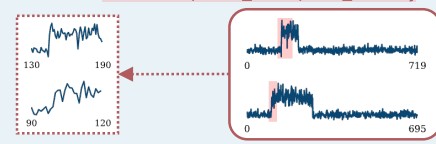

The comparison confirms that while both series exhibit a surge pattern, the duration is significantly shorter in the updated series. The original surge lasts approximately 31 hours, while the updated surge lasts only 4 hours. Is there any possibility I misidentified the start and end points? Let me check a slightly wider window to ensure I didn't miss any part of the surge.

This broader comparison reinforces my initial findings. The surge in the original series is clearly longer in duration than in the updated series.

The results show that the only significant surge in the updated series is the one I identified earlier, with no other comparable events. This confirms that the surge duration is indeed shorter in the updated series. **</think>**

**<answer>** \answer{The surge lasts for 7 days in the updated time series while it lasts 3 days in the original.}
**</answer>**

Figure 8: Case study of a comparison reasoning case.

In Figure 8, the model is required to carefully analyze the surges in the MTS and compare the differences in their lengths. It can be seen that ThinkTime automatically invoked the compare tool

to identify the starting points of the surges and finally analyzed the length difference between the two surges. This shows that ThinkTime can selectively invoke appropriate tools during iTCoT according to the context to obtain precise, detailed information.

## B   EVALUATION DETAILS

### B.1   EVALUATION TASKS

In this section, we provide a detailed introduction to each type of evaluation task. Specific cases can be found in the evaluation datasets provided in the supplementary material. Types of evaluation tasks are listed in Table 5.

Table 5: Details of different tasks in evaluation datasets

| Task | Task Type | Answer Type | # Questions | Data Source | Types of TS |
|---|---|---|---|---|---|
| Pattern Recognition | Reasoning | T/F | 130 | AIOPS, NAB, UCR | UTS, MTS |
| Numerical Judgment | Reasoning | T/F | 90 | AIOPS, NAB, UCR | UTS, MTS |
| Calculation | Reasoning | Num. | 85 | AIOPS, NAB | UTS, MTS |
| Causal | Reasoning | MC | 77 | AIOPS, NAB | UTS, MTS |
| Comparison | Reasoning | MC | 100 | MCQ2 | MTS |
| Trend | Alignment | Cate. & Num. | 41 | AIOPS, NAB, Oracle | UTS |
| Season | Alignment | Cate. & Num. | 37 | AIOPS, NAB, Oracle | UTS |
| Noise | Alignment | Cate. & Num. | 42 | AIOPS, NAB, Oracle | UTS |
| Local Fluctuation | Alignment | Cate. & Num. | 72 | AIOPS, NAB, Oracle | UTS |
| Correlation | Alignment | Cate. | 42 | Weather, Oracle | MTS |
| Cluster | Alignment | Cate. | 42 | Weather, Oracle | MTS |

### B.1.1   REASONING TASKS

Time series reasoning tasks require the LLMs to accurately identify features within the time series and perform reasoning by combining these features with textual descriptions. This places high demands on both the LLMs' multimodal recognition ability and their reasoning ability.

**Pattern Recognition.** Given a combination of attributes of a time series, the LLM determines whether the time series matches the description and provides a *yes/no* answer. For example, it may determine whether a time series satisfies the description "first rises, then falls, with no obvious noise".

**Numerical Judgment.** Given a combination of attributes of a time series with *numerical requirements*, the LLM determines whether the time series matches the description and provides a *yes/no* answer. For example, it may determine whether a time series satisfies the description "has an upward spike with amplitude larger than 5, with no downward convex with amplitude larger than 10".

**Calculation.** Reasoning tasks that require numerical answers can be counting tasks or calculation tasks. For example: "output the sum of the amplitudes of all upward spikes"; or "output the number of sudden increases with an amplitude greater than 5".

**Causal.** Based on time series features, choose the most likely event from multiple options. For example, given a time series representing memory usage, determine whether the program crashed or triggered GC.

### B.1.2   ALIGNMENT TASKS

Alignment tasks require the model to precisely output the category and numerical information of a feature (such as position or amplitude). They do not rely on reasoning ability and are only used to evaluate the multimodal alignment ability of the LLM.

**Trend.** Output the trend category (for example, increasing or decreasing) and the amplitude of change.

**Season.** Output whether periodicity exists, as well as the length and amplitude of each period.

**Noise.** Output whether large noise exists, as well as the standard deviation of the noise.

**Local Fluctuation.** Output the positions and amplitudes of all the local fluctuations, along with their types (choose from a given list).

**Correlation.** Determine whether the two time series have correlations in overall shape or local fluctuations. Output yes/no.

**Cluster.** Identify all time series that have correlations in overall shape or local fluctuations with a specified time series.

## B.2 EVALUATION METRICS

For reasoning tasks, we adopted different evaluation metrics according to their output formats. For T/F (True / False) and MC (Multiple Choice) type questions, we directly used string matching to evaluate accuracy. For numerical type questions, following Xie et al. (2025), we used relative accuracy $R_{num}$ as the evaluation metric:

$$R_{num} = \max\left(1.0 - \frac{|num_{ans} - num_{gt}|}{|num_{gt}|}, 0.0\right) \tag{2}$$

For alignment tasks, we used exactly the same evaluation method and code as in Xie et al. (2025). All evaluation metrics take values in the range of 0 to 1, with higher values indicating better performance.

## B.3 EVALUATION DATASETS

For reasoning tasks, we collected and annotated data from the following sources:

- **AIOPS** (Li et al., 2022). It contains multiple monitoring metric data collected from servers, along with anomaly annotations at failure times.
- **NAB** (Ahmad et al., 2017). It contains real-world timeseries collected from many scenarios, including social network, traffic, server, etc.
- **UCR** (Schmidl et al., 2022). It contains multiple time series collected from different scenarios, along with anomaly annotations.

For each of the datasets above, we randomly sampled time windows. With the assistance of an LLM, we generated captions for each time series, including a complete description of its trends, local fluctuations, periodicity, and other features. Based on this, we used the LLM to construct a series of reasoning questions and answers. We manually verified the correctness of each question to ensure the quality of the evaluation data. For the comparison reasoning task, we used the MCQ2 (Merrill et al., 2024) dataset, which contains time series and their corresponding questions and answers. Due to cost considerations, we randomly sampled 100 questions from it for evaluation.

## B.4 IMPLEMENTATION OF BASELINES

We implement 4 types of models in our evaluation: text-based, vision-based, agent-based, and TS-based. The implementation details are as follows:

- **Text-Based Methods.** The text-based baselines are implemented by converting time series into sequences of floating-point numbers, which are then represented in plain text. Outputs for OpenAI models, such as the GPT series, are obtained through the official API, while open-source models, such as the Qwen series, are deployed locally with vLLM to generate results.
- **Vision-Based Methods.** The vision-based baselines are implemented by transforming time series into line chart images using matplotlib. For multivariate time series, multiple subplots are created with aligned axes, and each curve is annotated with its corresponding metric name to preserve clarity.

- **Agent-Based Methods and Workflow-Based Methods.** The agent-based baselines are implemented with the ReAct framework, where a set of predefined tools can be invoked to support time series reasoning tasks. These tools cover operations including retrieving datapoint values, detecting anomalies, performing classification, and computing correlations across trends or fluctuations. We implement the agent-based methods following the same settings in ChatTS.

- **TS-Based Methods.** The TS-Based (Time Series MLLM) baseline is implemented with the official code and checkpoint released by the original authors. There is no need to convert the time series into other modalities, as it already supports native time series input.

For all implementation details about the baseline methods, please refer to the source code.

## C  TRAINING DETAILS

### C.1  RL REWARDS

The following rewards are used during the DAPO training of ThinkTime:

- **Accuracy Reward** ($R_{acc}$). The accuracy reward evaluates the correctness of the LLM output. For different categories of verifiable questions, distinct calculation methods are applied. The detailed definitions of these methods are provided in Section B.2.

- **Format Reward** ($R_{format}$). Following the original settings in Guo et al. (2025), the format reward is applied to ensure that the reasoning process is enclosed within the think tags (*i.e.*, <think> and </think>) and the final answer is enclosed within the answer tags (*i.e.*, <answer> and </answer>).

- **Thinking Length Reward** ($R_{len}$). The thinking length reward is defined as:

$$R_{len} = \mathbb{I}_{len(\text{thinking\_process})>L_{min}} \quad (3)$$

  where $\mathbb{I}$ denotes the binary function and $L_{min}$ is the minimum thinking length threshold. The thinking length reward is used to prevent the thinking process from collapsing. We set a minimum value of 1500 characters (in fact, almost all normal thinking processes exceed this threshold) to ensure the stability of the training process.

We set $w_{acc} = 5.0, w_{format} = 1.0, w_{len} = 1.0$ as the weights for these rewards. We do not set a reward for the number of tool calls (as DeepEyes (Zheng et al., 2025) does), because we found that this leads to instability in the training process. In fact, for some simple tasks, such as determining the overall trend of a stable time series, the LLM can complete them without calling any tools.

### C.2  IMPLEMENTATION OF DAPO FOR ITCOT

**Rollout Formulation for iTCoT.** Inspired by the design of iMCoT in DeepEyes Zheng et al. (2025), the time series chain-of-thought is also modeled as a Markov Decision Process (MDP). In this setting, the state at each step consists of the generated text together with the time series observations obtained from tool interactions. These observations may include results from operations such as slicing or comparing subsequences. The state is updated step by step as new tokens and observations are added. The rollout continues until either a final answer is produced or the maximum step limit is reached. Only the text outputs are used for computing the optimization objective, while the observation tokens serve as auxiliary context.

Formally, the state at step $t$ can be expressed as

$$s_t = \{X_{\leq t}, O_{\leq t}\},$$

where $X_{\leq t}$ denotes the accumulated text tokens and $O_{\leq t}$ denotes the collected observation tokens from time series tools. Based on the current state $s_t$, the model produces the next output token or triggers a tool call. The new text or observation is appended to the sequence, updating the state for the following step. This rollout proceeds until an answer is produced or the predefined step limit is reached. For optimization, only the text outputs are included in the loss computation, while the observation tokens act as auxiliary context.

**Implementation** To implement the DAPO of iTCoT, we made extensive modifications to the GR-POTrainer in trl von Werra et al. (2020). First, we added multimodal support so that it can perform the DAPO of TS-MLLM. Then, we aligned it with support for multi-turn rollout, which is required by iTCoT. By invoking the generation process multiple times in each rollout and computing the loss at the end, we finally achieved the DAPO of iTCoT. The corresponding code has been uploaded to the supplementary material, and we plan to make it open source.

### C.3  TRAINING SETTINGS

We use ChatTS-14B[1] as our base model, which was fine-tuned from Qwen2.5-14B-Instruct[2]. We conduct Warm-Up SFT training with LLaMA-Factory Zheng et al. (2024), using a learning rate of 1e-5, a batch size of 256, and 200 training steps. In the RL stage, we train with the modified TRL von Werra et al. (2020), setting the learning rate to 2e-6, the batch size to 32, and rollout_n to 8, for a total of 300 steps. We use the same model structure and settings as in ChatTS-14B. These step numbers were determined based on our experience. For detailed training procedures, please refer to the training code in the supplementary material.

## D  TRAINING DATASETS

### D.1  SYNTHETIC TIME SERIES GENERATOR

We use the **synthetic time series generator** from ChatTS (Xie et al., 2025) to generate all the time series in the training datasets. In the generator, all the attributes in the time series (including the trend, noise, seasonality, and the local fluctuations) are stored in a *attribute pool*. Captions of time series are constructed using a series of templates according to the attribute pool.

The time series generator defines an All Attribute Set that covers a broad spectrum of time series properties. Seasonal attributes include no periodic fluctuation, sinusoidal periodic fluctuation, square periodic fluctuation, and triangular periodic fluctuation. Trend attributes consist of decreasing, increasing, and steady behaviors. Frequency attributes distinguish between high and low frequency patterns, while noise attributes specify either noisy signals or signals with almost no noise.

In addition, the generator incorporates a diverse set of local fluctuation attributes: `shake`, `upward spike`, `downward spike`, `continuous upward spike`, `continuous downward spike`, `upward convex`, `downward convex`, `sudden increase`, `sudden decrease`, `rapid rise followed by slow decline`, `slow rise followed by rapid decline`, `rapid decline followed by slow rise`, `slow decline followed by rapid rise`, `decrease after upward spike`, `increase after downward spike`, `increase after upward spike`, `decrease after downward spike`, `wide upward spike`, `wide downward spike`. Multiple attributes can be combined within the same time series. Further details of the implementation are provided in the source code.

### D.2  WARM-UP SFT DATASETS

First, we generate a set of alignment datasets with the synthetic time series generator introduced in Section D.1. The generated datasets contain only `question` and `answer`, without any CoT part. Then, we use LLMs to generate the iTCoT parts (thinking text and tool call) based on the given questions and answers with task-specific prompts. Finally, we parse all the tool calls in the generated text and add the tool responses accordingly to get the final dataset for Warm-Up SFT. We use 22,600 samples in our Warm-Up SFT stage.

---

[1]https://huggingface.co/bytedance-research/ChatTS-14B
[2]https://huggingface.co/Qwen/Qwen2.5-14B-Instruct

### D.2.1 PROMPTS FOR TOOL USE

Available Tools:
- get_timeseries_slice: Retrieves a slice of time series data for detailed analysis
- Parameters: metric_name (choose from {', '.join(metrics)}), start (integer, >= 0), end (integer, < {timeseries_length})
- Returns: Time series values, statistics, and visualization for the specified range
- **Flexible Slice Length**: Adapt window size to purpose - 8-16 for focused inspection, 16-64 for standard analysis, larger for broad overview

- compare_timeseries_slice: Compares two time series slices for comparative analysis
- Parameters: metric_name_1, start_1, end_1, metric_name_2, start_2, end_2 (all integers >= 0, ends < {timeseries_length})
- Returns: Values and comparative statistics for both slices
- **Flexible Slice Length**: Adapt window size to purpose - smaller windows for specific pattern comparison, larger for overall trend comparison

Time series length: {timeseries_length}
**Strategic windowing**: Choose window sizes and positions based on your analytical goals - vary between focused inspection and broad analysis

### D.2.2 PROMPTS FOR LOCAL FLUCTUATION-RELATED iTCoT GENERATION

TASK-SPECIFIC GUIDANCE (Local Fluctuation Analysis):
- Focus on identifying and analyzing ALL local fluctuations (spikes, dips, convex/concave patterns)
- Use get_timeseries_slice to examine different segments for detailed local analysis
- **Flexible call count**: Use 2-4 tools, but adapt based on complexity - fewer for simple patterns, more for complex fluctuation distributions
- **Adaptive windowing**: Start from different time points (beginning/middle/end), use varied window sizes (16-64 points), sometimes target specific visible patterns
- Pay attention to amplitude, duration, and timing of fluctuations
- **Varied analytical approaches**: Sometimes scan systematically, other times target suspected fluctuation zones; occasionally start with broad overview then zoom in
- Use phrases like "Let me examine this spike more closely" or "I should check other segments for similar patterns"

Requirements:
- **Flexible tool usage**: Use 2-4 get_timeseries_slice calls, varying window positions and sizes based on observed patterns
- **Adaptive slice strategy**: Choose window sizes (16-64 points) and positions strategically - sometimes systematic scanning, other times targeted investigation
- Include self-reflection like noticing empty/flat slices and then correcting with a better slice
- **Varied reflection patterns**: Sometimes question window choice, other times doubt pattern interpretation, occasionally reconsider analytical approach
- Focus analysis on local characteristics, not overall trends
- Examine multiple time segments to identify all fluctuations
- Do not mention "context" - act as if you only see the timeseries data

### D.2.3 PROMPTS FOR TREND-RELATED iTCoT GENERATION

TASK-SPECIFIC GUIDANCE (Trend Analysis):
- Focus on overall increasing/decreasing/stable patterns across the entire timeseries
- NO TOOL CALLS needed - analyze the overall pattern without detailed segmentation
- Look at the general direction and long-term behavior
- Consider slope, monotonicity, and overall trajectory

Requirements:
- DO NOT use any tool calls for trend analysis
- Use self-reflection on ambiguous global patterns, then resolve it without tools
- Focus on overall direction and long-term patterns
- Use phrases like "Looking at the overall pattern..." or "The general trend shows..."
- Do not mention "context" - act as if you only see the timeseries data

### D.2.4 PROMPTS FOR SEASON-RELATED iTCoT GENERATION

TASK-SPECIFIC GUIDANCE (Seasonal Pattern Analysis):
- Identify repeating patterns and periodic behaviors
- Use compare_timeseries_slice to compare different periods of the same timeseries
- Look for cyclical patterns, regular intervals, and repeated structures
- **Flexible approach**: Use 1-3 tool calls - sometimes start with suspected period, other times explore different interval hypotheses
- **Varied windowing**: Experiment with different period lengths and alignment strategies based on visual observations

Requirements:
- **Adaptive tool usage**: Use 1-3 compare_timeseries_slice calls, varying period selection strategy
- **Flexible alignment**: Sometimes align periods systematically, other times test hypotheses about cycle length
- Allow initial comparisons to be inconclusive and then correct with better-aligned periods or different intervals
- Look for repeating patterns and cyclical behaviors
- **Varied reflection**: Question period selection, doubt alignment accuracy, or reconsider cycle length hypotheses
- Include self-reflection like "Let me compare this period with an earlier one..."
- Focus on periodicity and seasonal characteristics
- Do not mention "context"

### D.2.5 PROMPTS FOR NOISE-RELATED iTCoT GENERATION

TASK-SPECIFIC GUIDANCE (Noise Analysis):
- Focus on random variations, minor fluctuations, and statistical properties
- Avoid analysis of periods with local fluctuations; correct if the chosen slice contains events
- Use get_timeseries_slice to examine different segments for noise characteristics
- **Adaptive windowing**: Use varied window sizes (16-64 points) and positions - sometimes multiple small windows for comparison, other times fewer larger samples
- **Strategic selection**: Target visually calm regions, but occasionally check eventful areas for comparison before correcting
- Analyze variance, standard deviation, and random components
- **Flexible approach**: Use 1-4 tool calls depending on noise complexity and distribution across the series

Requirements:
- **Flexible tool usage**: Use 1-4 get_timeseries_slice calls, adapting to noise distribution pattern
- **Strategic window selection**: Vary sizes and positions, sometimes comparing calm vs eventful regions before focusing on appropriate areas
- If a slice contains strong local events, acknowledge and switch to a calmer slice
- Focus on statistical properties and minor fluctuations
- **Varied reflection approaches**: Sometimes question window choice, other times doubt statistical interpretation
- Do not mention "context"

### D.2.6 PROMPTS FOR LOCAL CORRELATION-RELATED iTCoT GENERATION

TASK-SPECIFIC GUIDANCE (Local Fluctuation Correlation):
- Analyze correlation of local fluctuations between two timeseries (position-based)
- **Flexible workflow**: Sometimes start with single-series analysis then compare, other times jump directly to comparison, occasionally use multiple comparison windows
- **Adaptive windowing**: Use varied approaches - synchronized windows, offset windows for lag analysis, or different window sizes for different aspects
- Allow initial misaligned windows and then correct to synchronized ones, or vice versa
- **Variable tool count**: Use 2-5 tool calls based on correlation complexity and temporal patterns

Requirements:
- **Flexible approach**: Use 2-5 tool calls with varied strategies - sometimes systematic single-then-compare, other times direct multi-series comparison
- **Adaptive correction**: If initial comparisons show weak alignment, vary the correction approach - adjust timing, window size, or analytical focus
- **Varied reflection**: Question synchronization assumptions, doubt window choices, or reconsider correlation methodology
- Focus on timing-based synchronization and positional correlation
- Do not mention "context"

### D.2.7 PROMPTS FOR SHAPE CORRELATION-RELATED iTCoT GENERATION

TASK-SPECIFIC GUIDANCE (Shape/Trend Correlation):
- Analyze overall shape and trend correlation between two timeseries
- Use ONLY ONE compare_timeseries_slice call for the entire timeseries comparison
- Reflect on whether one full-series comparison could hide phase-specific differences, but keep only one call

Requirements:
- Use EXACTLY ONE compare_timeseries_slice call for the full timeseries comparison
- Include reflective doubt but resolve with a clear global conclusion
- Focus on overall shapes and trends
- Do not mention "context"

### D.2.8 Prompts for Shape Cluster-Related iTCoT Generation

TASK-SPECIFIC GUIDANCE (Shape-based Clustering):
- Cluster timeseries based on overall shape and trend similarities
- Use compare_timeseries_slice to compare overall shapes between different series
- At most one comparison can be admitted as inconclusive; then correct
- Call tools 2-3 times

Requirements:
- Use 2-3 compare_timeseries_slice calls
- Focus on global shape similarities for clustering
- Allow one inconclusive compare then a corrected one
- Do not mention "context"

### D.3 RL Datasets

The RL datasets are divided into *alignment-related* and *reasoning-related* parts for RLVR with DAPO. For the alignment-related dataset, we directly reorganized the questions and answers from the WarmUp SFT dataset into a verifiable format (with a fixed answer format that can be parsed and evaluated by rule-based methods to compute accuracy). For the reasoning-related dataset, we used synthetic time series and their corresponding generated descriptions, and employed LLMs to generate different categories of questions. We use 30,000 samples in our RL stage, including 20,000 reasoning samples and 10,000 alignment samples.

### D.3.1 Prompts for Pattern-Recognition Dataset Generation

You are asked to design reasoning-style pattern recognition questions for time series. Each question must be based only on the provided attribute description and the corresponding time series.

**Given Attributes (do not reveal them in the question):**
{caption}

**Requirements:**
1. Use only the given attributes: *trend, local fluctuations (e.g., upward spike, convex, increase), seasonality, noise.*
2. The question must describe a **specific combination** of attributes and ask whether it exists in the given time series.
3. Do not reveal or hint at the ground-truth label in the question. The question must only describe the pattern conditions, not whether they are satisfied.
4. The answer must be strictly 'yes' or 'no'.
5. Output format must be valid JSON list, with different questions and answers, each item with fields '"question"' and '"answer"'.
6. Ensure the questions are diverse and challenging, which requires deep understanding of time series patterns. Avoid simple or trivial questions.

**Example:**
- **Attributes:** trend = upward, local fluctuations = upward spike, seasonality = none, noise = low
- **Output:**
json
{{
"question": "Does the time series contain an upward spike followed by a continuous upward trend?",
"answer": "yes"
}},...

### D.3.2 Prompts for Numerical-Judgment Dataset Generation

You are asked to design reasoning-style numerical judgment questions for time series in realistic business scenarios. Each question must be based only on the provided attribute description and the corresponding time series.

**Given Metric Name:** {metric_name}
**Given Attributes (do not reveal them in the question):**
{caption}

**Requirements:**
1. Consider only numerical attributes: *local fluctuation amplitude, seasonal amplitude, seasonal period, max and min values*. Completely ignore noise-related attributes, trend amplitude.
2. Create scenario-based questions with business context related to the metric type:
- **Threshold-based judgment:** Use realistic business thresholds and SLA requirements. The threshold must **not** be too close to the true value. For example, if local fluctuation amplitude = 31.01, set threshold = 20 (safe margin), not 30.
- **Comparison judgment:** Compare different attribute values in business contexts (e.g., performance vs capacity, peak vs baseline). Avoid cases where the two values are nearly equal.

3. The question must not leak the ground-truth values. Only describe realistic business conditions and scenarios.

4. The answer must be strictly 'yes' or 'no'. Questions should involve realistic monitoring scenarios, SLA compliance, alert conditions, etc.

5. Output format must be a valid JSON list with exactly 5 items, each item having '"question"' and '"answer"'.

6. Ensure the 5 questions are diverse and challenging, covering different business scenarios like performance monitoring, capacity planning, anomaly detection, SLA compliance, etc.

7. (VERY IMPORTANT) Make sure that the threshold is NOT close to the true value. There should be a clear gap between the threshold and the actual value to avoid ambiguity.

**Example:**
- **Metric:** CPU Utilization, **Attributes:** local fluctuation amplitude = 31.01, seasonal amplitude = 8.0, seasonal period = 20, max value = 40, min value = -10
- **Output:**
json
{{
"question": "In a production environment, CPU utilization spikes exceeding 20.0 above baseline are considered critical alerts that require immediate attention. Based on the observed fluctuation patterns, would this system trigger any critical alerts?",
"answer": "yes"
}}, ...

### D.3.3 PROMPTS FOR CALCULATION DATASET GENERATION

You are asked to design reasoning-style calculation questions for time series in realistic business and operational scenarios. Each question must be based only on the provided attribute description and the corresponding time series.

**Given Metric Name:** {metric_name}
**Given Attributes (do not reveal them in the question):**
{caption}

**Requirements:**
1. Focus on numerical attributes such as *local fluctuation amplitude, local fluctuation positions, count of local fluctuations, seasonal amplitude, seasonal period, max value, min value* and other meaningful values derived from them. Do not consider noise, trend-related amplitude, start and end values.
2. Create business-scenario calculation questions requiring reasoning or counting, with the final answer being a **single number**. Questions should be easy to understand and clear.
3. The question must not reveal the ground-truth numerical values. Only describe the business context and calculation methodology clearly.
4. The answer must be a single number without units or extra text.
5. Clearly state if the answer should be the absolute value or the original value (which can be negative).
6. Output format must be a valid JSON list with exactly 5 items, each item having '"question"' and '"answer"'.
7. Ensure the 5 questions are diverse and cover different business scenarios: financial analysis, performance monitoring, capacity planning, compliance checking, and operational analytics.
8. (VERY IMPORTANT) If the question is threshold-related, make sure that the threshold is **NOT** close to the true value. There should be a clear gap between the threshold and the actual value to avoid ambiguity.

**Example:**
- **Metric:** Memory Usage (GB), **Attributes:** trend amplitude = 12, spike 1 amplitude = 30 and pos = 123, spike 2 amplitude = 40 and pos = 54, spike 3 amplitude = 10 and pos = 201, seasonal amplitude = 8, seasonal period = 20 max value = 85, min value = 15
- **Output:**
json
{{
"question": "In incident management, each memory spike above normal operations with an amplitude of more than 15 triggers an alert. How many alert incidents would the monitoring system generate based on the observed spike patterns?",
"answer": "2"
}}, ...

### D.3.4 PROMPTS FOR CAUSAL DATASET GENERATION

You are asked to design reasoning-style causal multiple-choice questions for time series. Each question must be based on the provided attribute description and the metric name.

**Given Metric Name (use it in the question context):**
{metric_name}

**Given Attributes (do not reveal them in the question):**
{caption}

**Requirements:**
1. Construct a realistic scenario where the given attributes could arise in the {metric_name} time series.
2. Each question must include exactly 4 options (A, B, C, D) embedded directly in the question text. Only one option must be

```
correct.
3. Wrong options must be plausible but not consistent with the given attributes.
4. The question must require reasoning based on the attributes and the metric's real-world meaning, not just surface matching.
5. The difficulty should be non-trivial, requiring causal inference and detailed analysis of the timeseries themselves. The questions
should be misleading and hard to answer.
6. Output format must be a valid JSON list with exactly 5 items. Each item should have '"question"' and '"answer"'.
7. '"answer"' should be the correct option label (e.g., '"A"').

**Example:**
- **Attributes:** trend = upward, local fluctuation = upward spikes, seasonality = none, noise = low
- **Metric Name:** CPU Utilization
- **Output:**
json
{{
"question": "Which of the following events most likely caused the observed pattern in CPU Utilization, showing a general upward
trend with occasional sharp spikes? A) A gradual rollout of a new background data-processing job, with occasional batch tasks
triggering sharp increases. B) A temporary network outage that completely stops CPU usage. C) Stable user traffic with no
significant changes in workload. D) A constant low-level background process running without interruptions.",
"answer": "A"
}}, ...
```

## E  DISCUSSION

**Limitation.** While ThinkTime demonstrates strong improvements in both reasoning and alignment, several limitations remain. First, our training pipeline relies heavily on synthetic data for constructing interleaved time series CoT and RLVR datasets. Although these datasets are carefully designed and verified, they cannot fully capture the diversity and complexity of real-world time series, which may limit the generalization of the model to unseen domains. Second, our evaluation benchmarks, although broad, are still constrained by the availability of existing datasets. The absence of large-scale standardized benchmarks for time series reasoning makes it difficult to measure progress in a fully consistent way. Finally, the current model is based on a single base model, and the scalability of our approach to larger or different base models has not been systematically examined.

**Future Work.** Future research can address these limitations in several directions. Expanding real-world datasets for alignment and reasoning tasks is an important step toward improving the robustness and generalization of time series multimodal LLMs. Another promising direction is to design standardized reasoning benchmarks that cover a wide range of time series tasks, which would provide a more reliable basis for evaluation and comparison. In addition, exploring more advanced reinforcement learning algorithms and richer reward functions may further enhance training stability and reasoning capability. Extending the framework to other backbones and scaling it to larger model sizes could also test the adaptability of our approach. Finally, integrating domain-specific knowledge and applications, such as forecasting or anomaly diagnosis, may broaden the practical impact of multimodal deep thinking with time series.

## F  USE OF LARGE LANGUAGE MODELS

In the writing of this paper, LLMs were used for translation and text polishing. LLMs were also applied in the data generation process of this paper. The detailed generation process can be found in the relevant sections above.

