# OpenReview forum: "Thinking with Time Series: Interleaved Deep Thinking for Enhanced Time Series Reasoning"
_ICLR.cc/2026/Conference — Submitted to ICLR 2026_

### Official Review · Reviewer_it4P · 2025-10-28

**Soundness:** 2
**Presentation:** 2
**Contribution:** 2
**Rating:** 2
**Confidence:** 5

**Summary:**

The paper introduces ThinkTime, a framework for time-series reasoning via interleaved Chain-of-Thought (iTCoT) using simple slice and compare tools. Trained on synthetic ChatTS data with RLVR, it shows structured reasoning but limited real deep thinking.

**Strengths:**

The experiments are well-organized and cover multiple benchmarks, showing consistent improvements over baseline time-series LLMs. The evaluation includes both alignment and reasoning tasks, demonstrating that the proposed iTCoT approach.

**Weaknesses:**

1. The tool design is too simple
Only two operations, `slice` and `compare`, are provided, which makes it difficult to capture complex relationships. There is a lack of deeper analytical tool design, and I think there is a large gap between this and what is claimed as *DEEP THINKING*.

2. The reasoning is more like “description”
The model mostly observes the figures and gives language summaries, rather than performing real statistical or causal reasoning.
Furthermore, in the DAPO training there is no supervision information for complex reasoning chains, and most prompts are just template-like instructions.
The RLVR reward function also only considers format, accuracy, and length, without taking reasoning quality or the rationality of tool usage into account.

3. The training data come from ChatTS
Although an additional CoT part was added, after checking Appendix D, the process is overly regularized and lacks diversity of real thinking. Most are rather general instructions. I doubt whether the current model can truly follow these instructions, or is merely producing hallucinatory outputs.

**Questions:**

Please see the weaknesses discussed above.

---

> ### Author Response · Authors · 2025-11-28
>
> We sincerely thank the reviewer for the valuable feedback. Please find our responses below.
>
> **Q1: Are only two operations (slice and compare) sufficient, and is there a large gap between this and real "deep thinking"?**
>
> A1: Thank you for your question. To clarify our core contribution: "thinking with time series" means interleaving multimodal tokens (time series observations) into the reasoning process to enhance accuracy and reduce hallucination, not building a comprehensive tool library. Our motivation is that time series have varying lengths and scales, and feeding entire sequences into LLMs causes substantial detail loss, preventing precise reasoning about time series features. Slice and compare serve as implementation mechanisms that require no additional algorithmic implementation and enable reasoning about any temporal aspect at appropriate scales through per-slice normalization. In our paper, "deep thinking" refers to iteratively examining details, updating understanding, and refining conclusions based on the actual observations rather than solving difficult problems that require multi-step reasoning. This is similar to the reasoning process of other multimodal LLMs, which also include extensive, detailed descriptions of multimodal features (e.g., zoom-in an image to better understand the details). To avoid misunderstanding, we will avoid using the ambiguous "deep thinking" term to describe the iTCoT process.
>
> **Q2: The reasoning is more like description rather than actual statistical or causal reasoning. The RLVR reward function is not adequate to capture reasoning quality**
>
> A2: Thank you for raising this important point. The ThinkTime model is actually doing multimodal reasoning about the time series details rather than simple descriptive narration. As shown in Appendix A, the model not only identifies the time series patterns but also forms hypotheses and verifies them through selective slicing. For example, in Figure 5, the model first detects a seasonal pattern, then actively checks amplitudes in multiple segments before concluding whether the trend doubles, demonstrating analytical reasoning rather than simple summarization.
>
> Regarding the reward design, our reward does not explicitly evaluate each intermediate reasoning step. This choice follows the DAPO framework and is consistent with prior RLVR approaches, where the final correctness of the answer serves as a reliable and scalable supervision signal. In time-series reasoning tasks, correct conclusions typically require several grounded observations made through interleaved tool calls. Therefore, the accuracy reward indirectly reinforces analytical reasoning behaviors, as evidenced by the improved intermediate observations shown in Appendix A.
>
> We also tried adding a penalty on the number of tool calls to enforce more efficient reasoning. However, we found that introducing such penalties during early training constrained exploration and significantly harmed the model’s ability to learn how and when to call tools effectively. We therefore use the WarmUp SFT with a task-specific CoT process to teach the reasoning format, while RL focuses on refining the decision to retrieve evidence that supports correct conclusions. The performance gains across all task categories confirm that the reward design successfully promotes deeper reasoning rather than surface-level description.

---

> ### Author Response · Authors · 2025-11-28
>
> **Q3: Training data use structured templates and may not reflect real or diverse reasoning. Can the model truly follow instructions or is it hallucinating?**
>
> A3: Thank you for this careful observation. We agree that the warm-up SFT data use a structured iTCoT format. This design has a clear goal. It teaches the model how to interleave the tool calls with text in a stable way, so that it can learn the basic pattern of "Thinking with Time Series". The templates control the outer structure, but the actual content of each generated CoT depends on the generated time series attributes (e.g., trend, season, local fluctuations, correlation and noise). These attributes vary widely, so the model sees many different shapes and combinations even though the instruction style is similar, which is crucial for generalization.
> At the same time, the model is not trained only on these alignment style instructions. In the RLVR stage, we construct reasoning tasks where iTCoT is not required and answers are verifiable by numerical values or discrete labels. The model must combine the question with time series slices to give correct outputs on pattern, numerical, calculation, and causal tasks. The main results in Table 1 and the ablation in Table 3 show that ThinkTime improves accuracy over ChatTS and over the without Tool Use variant across almost all reasoning categories.  If the model is only repeating generic instructions or hallucinating, such consistent gains are not expected, especially on numerical judgment and calculation, where small mistakes are directly reflected in accuracy.
>
> Furthermore, the case studies in Appendix A illustrate that the model follows and internalizes the interleaved instructions. For instance, in the comparison example on seasonality, the model first reads the context, then calls compare on two segments of the original and updated series, checks the amplitudes in each segment, and only then concludes that the seasonal peak is doubled.  In another example, the model checks different parts of a multivariate time series to list correlated fluctuations. In these reasoning traces, the model does not simply repeat the templates. It selects specific intervals, inspects the returned slices, and adjusts its conclusion based on what it observes from the model itself.

---

### Official Review · Reviewer_L55B · 2025-10-29

**Soundness:** 3
**Presentation:** 3
**Contribution:** 2
**Rating:** 4
**Confidence:** 4

**Summary:**

The paper proposes ThinkTime, a time-series multimodal LLM (TS-MLLM) that introduces interleaved Time-series Chain-of-Thought (iTCoT) with two tool operations—slice and compare—to enable deep, step-wise reasoning over multivariate time series. The model uses a two-stage training pipeline: Warm-Up SFT on iTCoT alignment data (largely synthetic) and RL with verifiable rewards (RLVR) using DAPO/TRL to improve tool use and reasoning trajectories. The authors curate evaluation suites spanning five reasoning categories (pattern, numerical, calculation, causal, comparison) and six alignment subtasks.

**Strengths:**

The paper makes a clear, original step toward time-series deep reasoning by operationalizing interleaved CoT with slice/compare tools and by delivering a complete training recipe. The empirical coverage across reasoning and alignment tasks is broad and convincing, with informative ablations and robustness analyses. ThinkTime shows substantial, consistent improvements over strong baselines and retains good alignment, suggesting the approach is both effective and practical.

**Weaknesses:**

1. The slice and compare operators are intuitive but remain informally defined.

2. there is no analysis of when/why iTCoT should reduce reasoning error vs. text-only CoT, nor bounds on over-slicing or mis-alignment risks in multivariate settings, consider add sensitivity analyses to slicing granularity and normalization choices.

3. Warm-Up SFT and much of RLVR rely on synthetic data. While results on real benchmarks are positive, the work lacks systematic transfer diagnostics, e.g., feature distribution distances, task-wise error taxonomy, and negative cases where synthetic priors mislead real-world reasoning. Provide dataset shift measurements and per-category failure analyses.

4. Little analysis of failure modes, such as false causal attributions, period misestimation under nonstationarity. Given iTCoT’s autonomy to call tools, a section on unsafe or privacy-sensitive tool uses would be valuable. Please add fine-grained error taxonomy and safeguards.

5. The approach focuses on regular numeric time series; extensions to event and irregular sampling, missingness, or exogenous interventions are mentioned implicitly via tools but not validated. Please consider add tests on missing data and irregular timestamps.

**Questions:**

Please see the above weaknesses, and the following:

1. How do you impose or learn a tool-call budget to prevent over-slicing? Is there a learned stopping policy or hard cap per turn/task?

2. Do you quantify distribution shift between synthetic and real data?

3. Beyond series length/number, have you tested robustness under noise, distribution shifts, or missingness?

4. The “causal” category is intriguing. Are these counterfactual or correlational diagnostics? What guarantees exist against spurious causal claims?

5. Please add ablations on reward weights, introduce penalties for degenerate tool use, and report compute and latency-accuracy trade-offs.

6. Please share ablations for $w_{\mathrm{acc}},\, w_{\mathrm{format}},\, w_{\mathrm{len}}$ and whether an efficiency/latency term or redundancy penalty improves behavior.

---

> ### Author Response · Authors · 2025-11-28
>
> We sincerely thank the reviewer for the valuable feedback. Please find our responses below.
>
> **Q1: How do you impose or learn a tool-call budget to prevent over-slicing? Is there a learned stopping policy or hard cap per turn/task?**
>
> A1: We tried using hard limits on tool call numbers to prevent too many calls, but found this did not improve model performance. The reason is that during DAPO, the model can decide by itself when to use tools, guided by the accuracy reward to optimize end-to-end correctness. Also, different tasks need different numbers and patterns of tool calls, making it hard to apply the same rules during RL. Instead, we handle potential over-slicing mainly during warmup SFT data construction by considering what each task needs, such as limiting slice window lengths and tool call counts in training examples. Figure 4a shows that this approach leads to natural adaptation where tool usage varies based on task complexity.
>
> **Q2: Do you quantify the distribution shift between synthetic and real data?**
>
> A2: Our evaluation mainly uses real-world time series from AIOPS, NAB, and UCR datasets for reasoning tasks. We believe the strong performance on real-world evaluation (Tables 1 and 2) has already shown successful transfer from synthetic training data. However, we do not yet know the method to quantify the distribution shift between time series datasets, so we did not directly verify the distribution shift in the paper.
>
> **Q3: Beyond series length/number, have you tested robustness under noise, distribution shifts, or missingness?**
>
> A3: Our alignment tasks include testing under different noise levels. Distribution shifts (i.e., concept drift) are also covered in our sudden increase/decrease category in our evaluation of local fluctuations. However, our model does not have a special design for missing points, as handling missing values is typically part of the preprocessing step before analysis. We will clarify this and discuss it as a preprocessing consideration rather than a model-level limitation.
>
> **Q4: The "causal" category is intriguing. Are these counterfactual or correlational diagnostics? What guarantees exist against spurious causal claims?**
>
> A4: Our causal reasoning tasks involve giving several options that represent different events, and based on the time series, the model needs to determine which event is most likely to have happened. These tasks do not involve counterfactual or correlational diagnostics in the time series domain. They are more like event-based reasoning. We only need to make sure that the features described in the given options can actually cause the physical results reflected in the time series. We use manual checking to ensure the correctness of this process.
>
> **Q5&Q6: Please add ablations on reward weights (wacc, wformat, wlen), introduce penalties for degenerate tool use, and report compute and latency-accuracy trade-offs. Also share whether an efficiency/latency term or redundancy penalty improves behavior.**
>
> A5&A6: Thank you for these valuable suggestions. We appreciate your thorough review and recognize that these ablation studies would provide additional insights into our design choices. However, given that each model training run requires several days to complete, we are unable to conduct these experiments within the rebuttal period. We believe these ablations, while informative, may not be critical to demonstrating the core contributions of our work which focus on the effectiveness of interleaved multimodal CoT for time series understanding. Our current reward setting was selected based on preliminary experiments showing balanced performance across accuracy and reasoning quality.

---

### Official Review · Reviewer_BP6D · 2025-10-30

**Soundness:** 2
**Presentation:** 2
**Contribution:** 2
**Rating:** 4
**Confidence:** 5

**Summary:**

The paper proposes ThinkTime, a multimodal large language model (TS-MLLM) for time-series reasoning that introduces Interleaved Time-series Chain-of-Thought (iTCoT) — a mechanism where reasoning steps are alternated with external “tool calls” such as slice and compare. Inspired by “Thinking with Images” (OpenAI, 2025), the model allows dynamic inspection of local temporal regions during reasoning. The training pipeline consists of two stages: (1) warm-up supervised fine-tuning (SFT) with synthetic iTCoT data, and (2) reinforcement learning with verifiable rewards (RLVR) under the DAPO framework. Experiments on 11 datasets show large gains over text, vision, agent, and previous TS-MLLM baselines (e.g., ChatTS-14B).

**Strengths:**

1. Novel conceptual extension: Introducing interleaved tool-based CoT for time-series reasoning is original and clearly differentiates from prior TS-MLLMs that only employ static CoT or fixed-window representations.

2. Well-structured system design: The paper provides an end-to-end pipeline (data construction → SFT → RL → evaluation) with a clear operational flow.

3. Comprehensive evaluation: Results include both reasoning and alignment benchmarks, with ablations on tool usage, RL, and workflow variants.

4. Readable presentation: Figures (1–4) and examples of tool calls make the paradigm understandable and reproducible.

**Weaknesses:**

1. The core assumption—that a reasoning paradigm proven effective for images (cropped regions and visual focus) directly generalizes to time-series—is not theoretically supported.

2. The contribution is largely an architectural composition (existing LLM + tool-call loop + RL). No new algorithmic component (e.g., reward shaping, CoT optimization, or reasoning trace modeling) is introduced. The reinforcement learning setup merely adapts DAPO with task-specific rewards but lacks theoretical or empirical insights into why RL improves reasoning in the temporal domain.

3. The experimental analysis is insufficient to substantiate the paper’s core claims. There is no ablation isolating the effects of the key slice and compare operations, baseline comparisons are potentially unfair due to inconsistent fine-tuning settings, and the tool-use evaluation lacks quantitative interpretability (e.g., precision, redundancy, or effectiveness). As a result, the reported performance gains cannot be confidently attributed to the proposed iTCoT mechanism.

**Questions:**

1. Logical Coherence and Transitions
The transition from the problem statement (“TS-MLLMs struggle with complex tasks”) to the core claim (“deep thinking is essential”) feels abrupt. The authors should explicitly articulate why “deep thinking” is necessary for time-series reasoning—e.g., because it allows iterative observation and temporal abstraction that single-pass reasoning cannot achieve.
2. Method and Contribution Clarity
The abstract presents too many methodological details at once, making it difficult to distinguish the core innovation. The authors should streamline the description, ensure consistent terminology, and clearly highlight the main contributions beyond prior TS-MLLMs.
3. Cross-Modal Generalization Assumption
The assumption that methods effective for image reasoning will directly apply to time-series reasoning lacks justification. The authors should clarify what aspects are transferable and provide supporting evidence.
4. Experimental Validation
How can the authors ensure that the reported improvements genuinely stem from the proposed iTCoT mechanism?
There is no ablation isolating the roles of slice and compare operations.
The fairness of baseline comparisons (fine-tuned vs. zero-shot models) is unclear.
The quantitative effectiveness of tool use (e.g., precision, redundancy, or correlation with accuracy) is not analyzed.
Providing these analyses or additional experiments could significantly strengthen the credibility of the results.
5. Minor Issue
The paper exhibits inconsistencies in capitalization and formatting, such as mixed usage of “Interleaved Time series Chain-of-Thought” vs. “Interleaved Time-Series Chain-of-Thought,” inconsistent reference styles (e.g., “OpenAI, c.” vs. “OpenAI (b)”), occasional mismatches between figure numbers and citations, and irregular quotation marks in JSON examples. These should be standardized for clarity and professionalism.

---

> ### Author Response · Authors · 2025-11-28
>
> We sincerely thank the reviewer for the valuable feedback. Please find our responses below.
>
> **Q1: Clarify the logical transition from "TS-MLLMs struggle" to "deep thinking is essential"**
>
> A1: Thank you very much for your valuable suggestion. We believe that the real reason why TS MLLMs need deep thinking is that the time series multimodal reasoning process must combine multimodal observations (which correspond to multimodal tokens) with understanding and thinking about the question (which correspond to text tokens). Therefore, this naturally leads to our iTCoT approach.
>
> **Q2: How do you support the assumption that reasoning paradigms proven for images directly generalize to time series?**
>
> A2: Our work does not assume direct generalization but draws inspiration from the underlying principle: complex multimodal reasoning benefits from iteratively incorporating multimodal observations. A key motivation is that time series have varying lengths and scales, and inputting the entire time series into LLMs causes large detail loss (see Figure 1). Both images and time series require reasoning about details at different scales, but we explicitly recognize unique temporal properties. Therefore, slice with per-slice normalization for temporal scale variations and compare for correlation analysis. These operations require no additional algorithmic implementation in retrieving details.
>
> **Q3: The core contribution and novelty exist beyond the architectural composition of existing components.**
>
> A3: Thank you for your question. Our contribution enables a new capability for TS-MLLM: interleaved multimodal reasoning for time series. This addresses the fundamental problem that feeding entire time series into LLMs loses critical details needed for precise reasoning. We want to clarify that our contribution is to present a feasible training process and to successfully verify that iTCoT is effective for time series reasoning tasks. This required several technical solutions. First, we developed the first DAPO training framework for time series MLLMs with interleaved reasoning, and extended TRL to support multi-turn rollouts with time series observations as external tokens. This is a new training pipeline for TS-MLLMs that has not been proposed or implemented in existing work. Furthermore, our task-specific data generation addresses that existing LLMs cannot produce meaningful interleaved time series reasoning, requiring the systematic construction of trajectories with appropriately placed observations.
>
> **Q4: No ablation studies of iTCoT and the baseline settings are unfair.**
>
> A4: Tables 3 and 4 provide ablation studies of tool use and reinforcement learning. The "w/o Tool Use" variant shows the ablation study of iTCoT, which uses identical questions and answers but text-only CoT.
> Regarding baseline fairness, we follow the common settings in TS-MLLM research. We want to emphasize that our contribution lies in the reasoning ability of our model, not in the base model itself. Therefore, this ability is independent of the base model. We only need to use a small base model, such as Qwen2.5-14B, and train it with our method to achieve performance that surpasses a larger model such as GPT 5. This is enough to verify that our training process is effective. Other models can all be trained using the same method.
>
> **Q5: Formatting issues and too many methodological details in the abstract.**
>
> A5: We sincerely thank the reviewer for pointing out these issues. We will correct the formatting problems and revise the abstract in the revised version.

---

### Official Review · Reviewer_aYSq · 2025-10-31

**Soundness:** 1
**Presentation:** 2
**Contribution:** 2
**Rating:** 4
**Confidence:** 3

**Summary:**

his paper introduces ThinkTime, a time series multimodal large language model (TS-MLLM) developed to address the challenges of complex reasoning tasks. The model incorporates a method called Interleaved Time series Chain-of-Thought (iTCoT), which allows the reasoning process to be combined with tool calls.

The framework uses two main operations, slice and compare, to let the model examine specific segments of a time series and analyze correlations between different parts. ThinkTime is developed through a two-stage training process that begins with Supervised Fine-Tuning (SFT) to learn the iTCoT process, followed by Reinforcement Learning (RL) to enhance its reasoning and tool-use capabilities. The authors report that this approach leads to noticeable improvements in the model's performance on various reasoning and alignment tasks involving real-world time series data

**Strengths:**

* proposed a methodology that can use integrated "tools" that explore specific components  and comparisons between time-series. This "tool" based method is novel for the time-series space, and has the potential to more deeply understand and probe a given time-series. Due to this, a DAPO approach for agent RL learning is also new and interesting for this time-series MLLM space.
* code is publicly available upon submission

**Weaknesses:**

* the core focus of this paper is on "slice" and "compare" operations and yet, in my opinion, the paper does not sufficiently explain why these two operations are fundamental for reasoning for time-series. I agree that they are important, yes, but the only two things that are necessary? I am not sure. The ablations also do not break down the tool use between the two. the paper would benefit from further discussion as to why these two exact tools ideas are used.
* Further explanation on the agent based models are needed in order to understand what capabilities they have available to them to make sure this is a fair comparison, especially with them having significantly worse performance than the TS methods.
* A key contribution is "propose a comprehensive data pipeline to support iTCoT" but it seems that most of these details are relegated to the appendix, so I am struggling to understand the core technical contribution of this, other than specific prompting techniques.
* I am unclear on the evaluation. How the evaluation datasets are constructed is also relegated into the appendix, so it is difficult to understand how they evaluate reasoning + alignment specifically. Even after I look into the appendix "Based on this, we used the LLM to construct a series of reasoning questions and answers. We manually verified the correctness of each question to ensure
the quality of the evaluation data. " -> this is very unclear as to how exactly reasoning is tested among other ambiguity.
* workflow-based model is suddenly introduced in the ablation study, and it is not clearly explained, so it is difficult to contextualize the comparison and understand exactly how ICoT improves upon it.

**Questions:**

* When constructing the RLVR data and LLM-as-a-judge is used to verify QA quality. How do we know that the LLM-as-a-judge was sufficient?

---

> ### Author Response · Authors · 2025-11-28
>
> We sincerely thank the reviewer for the valuable feedback. Please find our responses below.
>
> **Q1: Why are slice and compare the only two fundamental operations needed for time series reasoning?**
>
> A1: We appreciate the reviewer's question. To clarify our core contribution: "thinking with time series" means interleaving the multimodal tokens of time series into the reasoning process to enhance accuracy and reduce hallucination, rather than building a comprehensive tool library. Our motivation is from a key observation: time series have varying lengths (number of time points) and scales (numerical amplitudes), and inputting them entirely into the LLM causes detail loss (as shown in Figure 1), preventing precise reasoning about time series features. To achieve precise reasoning, we found iTCoT provides a highly flexible solution. Slice and compare require no additional algorithmic implementation and can examine any aspect of the time series through per-slice normalization. These operations serve as sufficient primitives for incorporating the time series information into thinking, enabling the model to adaptively focus on relevant temporal regions at appropriate scales.
> As for more complex tools such as STL decomposition, anomaly detection, and classification, we believe that LLMs have already been well trained to accurately identify these time series features. From the results in Table 2, we can see that even an Agent that uses tools still finds it hard to surpass our model on the basic alignment task. Therefore, our work focuses on using basic tools to improve the upper bound of the model’s ability in accurate analysis.
>
> **Q2: What capabilities are available to agent-based models to ensure fair comparison?**
>
> A2: Thank you for your question. Following ChatTS, our agent baselines use their ReAct Agent implementation with comprehensive tools including value STL, anomaly detection, classification, and Pearson correlation.
>
> **Q3: What is the core technical contribution beyond prompting?**
>
> A3: Thank you for your question. Our data pipeline addresses a fundamental challenge: existing LLMs cannot generate high-quality interleaved multimodal reasoning because they lack understanding of when to incorporate time series observations into thinking. Our task-specific generation systematically constructs reasoning steps where observations are naturally integrated at appropriate points, teaching the semantics of multimodal thinking. Combined with our RLVR framework and the first DAPO implementation for time series MLLMs with interleaved reasoning, this enables models to learn when to incorporate multimodal observations.
>
> **Q4: How is reasoning evaluated, and how do you verify the LLM-as-a-judge for RLVR data quality?**
>
> A4: For reasoning evaluation, we employ exact matching for categorical questions, relative accuracy for numerical answers, and manual verification for all constructed questions.
> Each question is created from the synthetic time series with known ground-truth attributes. Therefore, the data quality of RLVR has been guaranteed, as all attributes are already known by the LLM. Furthermore, we use another LLM to filter obvious errors and ensure non-trivial questions, but all questions are grounded in verifiable synthetic attributes. These techniques together ensure the quality of the generated data.
>
> **Q5: What is the workflow-based model introduced in ablations, and how does iTCoT improve upon it?**
>
> A5: The workflow baseline prompts GPT-5 to call slice and compare externally, which is similar to traditional agent approaches where tool usage is achieved through prompting. Instead, iTCoT integrates observations directly into the continuous reasoning process, allowing the model to dynamically retrieve and normalize specific temporal regions as needed by itself. The ablation studies in Table 3 and Table 4 demonstrate that embedding multimodal thinking into the model provides fundamental improvements over external orchestration.

---

### Meta-Review · Area_Chair_K5CJ · 2026-01-05

**Summary:**

This paper introduces ThinkTime, a TS-MLLM that utilizes interleaved Chain-of-Thought (iTCoT) with slice and compare operations to enhance time series reasoning. Reviewers found the idea interesting and empirically promising, but raised consistent concerns about the rigor and validation of the work. Key issues include insufficient justification for the fundamental nature of slice and compare, overstatement of “deep thinking” without intermediate reasoning metrics, limited ablations that isolate the contributions of iTCoT and tool use, and reliance on synthetic or templated data with opaque evaluation procedures. While the rebuttal clarifies intentions and provides illustrative examples, it does not fully address the need for formal analysis, systematic diagnostics, or robust empirical evidence. Overall, the work is conceptually appealing but not convincingly validated, placing it at the borderline reject.

**Reviewer Concerns:**

1. Insufficient justification and formalization of slice and compare as fundamental operations
2. Weak evidence that iTCoT enables “deep thinking” rather than descriptive narration
3. Lack of convincing experimental isolation and ablation
4. Data pipeline and evaluation methodology lack transparency
5. Reinforcement learning contribution is seen as incremental and under-analyzed
6. Overgeneralization and limited scope of validation

See my elaboration below (Reviewer Scores).

**Reviewer Scores:**

The AC believes the reviewers would be partially but not fully satisfied with the rebuttal responses. The responses address several of R1’s questions at a high level, but they do not completely resolve the reviewer’s underlying concerns, especially those about rigor, clarity, and empirical isolation (e.g., justification of slice and compare, evaluation clarity, and reasoning verification). For R2, the responses clarify intent but do not provide the theoretical grounding, ablation rigor, or causal evidence as R2 explicitly requested. For R2's key critiques (e.g., incremental novelty, lack of principled RL insight, insufficient experimental isolation), they remain fundamentally unaddressed. Similarly, for R3, several key questions are addressed with explicit acknowledgments of missing analysis, without introducing any new evidence, diagnostics, or safeguards. For R4, the gap between slice/compare and full statistical or causal reasoning is admitted but not mitigated. Reward design still does not explicitly evaluate intermediate reasoning steps. Reliance on synthetic, templated training data remains a potential limitation. No quantitative error analysis, failure modes, or robustness tests are added.

Based on the above observation, the AC believes that the reviewers would likely remain negative about this paper.

---

### Decision · Program_Chairs · 2026-01-26

Reject